# A two-stage blending approach for merging multiple satellite precipitation estimates and rain gauge observations: An experiment in the northeastern Tibetan Plateau

Yingzhao Ma[1], Xun Sun[2,3], Haonan Chen[1,4], Yang Hong[5], Yinsheng Zhang[6,7]

[1]Colorado State University, Fort Collins, CO 80523, USA
[2]Key Laboratory of Geographic Information Science (Ministry of Education), East China Normal University, Shanghai 200241, China
[3]Columbia Water Center, Earth Institute, Columbia University, New York, NY 10027, USA
[4]NOAA/Physical Sciences Laboratory, Boulder, CO 80305, USA
[5]School of Civil Engineering and Environmental Science, University of Oklahoma, Norman, OK 73019, USA
[6]Key Laboratory of Tibetan Environment Changes and Land Surface Processes, Institute of Tibetan Plateau Research, Chinese Academy of Sciences, Beijing, 100101, China
[7]CAS Center for Excellence in Tibetan Plateau Earth Sciences, Beijing, 100101, China

*Correspondence to*: Xun Sun (xs2226@columbia.edu)

**Abstract.** Substantial biases exist in the satellite precipitation estimates (SPE) over complex terrain regions and it has always been a challenge to quantify and correct such biases. The combination of multiple SPE and rain gauge observations would be beneficial to improve the gridded precipitation estimates. In this study, a two-stage blending (TSB) approach is proposed, which firstly reduces the systematic errors of original SPE based on a Bayesian correction model, and then merges the bias-corrected SPE with a Bayesian weighting model. In the first stage, the gauge-based observations are assumed as a generalized regression function of SPE and terrain feature. In the second stage, the relative weights of bias-corrected SPE are calculated based on the associated performances with ground references. The proposed TSB method has the ability to exert benefits from the bias-corrected SPE in terms of higher performance, and mitigate negative impacts from the ones with lower quality. In addition, Bayesian analysis is applied in the two phases by specifying the prior distributions on model parameters, which enables to produce the posterior ensembles associated with their predictive uncertainties. The performance of the proposed TSB method is evaluated with independent validation data in the warm season of 2010-2014 in the northeastern Tibetan Plateau. Results show that the blended SPE is greatly improved compared to the original SPE, even in the heavy rainfall event. This study can be expanded as a data fusion framework in the development of high-quality precipitation product in any region of interest.

## 1 Introduction

High-quality precipitation data is fundamental to understand the regional and global hydrological processes. However, it is still difficult to acquire accurate precipitation information in the mountainous regions, e.g., Tibetan Plateau (TP), due to limited

ground sensors (Ma et al., 2015). The satellite sensors can provide precipitation estimates at a large scale (Hou et al., 2014), but performances of available satellite products vary among different retrieval methods and climate areas (Yong et al., 2015; Prat and Nelson, 2015; Ma et al., 2016). Thus, it is suggested to incorporate precipitation estimates from multiple sources into a fusion procedure with a full consideration of the strength of individual members and associated uncertainty.

Precipitation data fusion was initially reported by merging radar-gauge rainfall in the mid-1980s (Krajewski, 1987). The Global Precipitation Climatology Project (GPCP) was an earlier attempt for satellite-gauge data fusion, which adopted a mean bias correction method and an inverse-error-variance weighting approach to develop a monthly, 0.25° global precipitation dataset (Huffman et al., 1997). Another popular dataset, the Climate Prediction Center Merged Analysis of Precipitation (CMAP), included global monthly precipitation with a 2.5° x 2.5° spatial resolution for a 17-year period by merging gauges, satellites and reanalysis data using the maximum likelihood estimation method (Xie and Arkin, 1997). Since then, several blending approaches have been developed to generate gridded rainfall product with higher quality by merging gauge, radar and satellite observations (e.g., Li et al., 2015; Beck et al., 2017; Xie and Xiong, 2011; Yang et al., 2017; Baez-Villanueva et al., 2020). Overall, those fusion methods follow a general concept by eliminating biases in satellite/radar-based data and then merging the bias-corrected satellite/radar estimates with point-wise gauge observations. However, these efforts might be insufficient for quantifying the predicted data uncertainty. Some blended estimates are also partially polluted by the poorly performed individuals (Tang et al., 2018).

This paper develops a new data fusion method that enhances the quantitative modelling of individual error structures, prevents potential negative impacts from lower-quality members, and enables an explicit description of model's predictive uncertainty. In addition, a Bayesian concept for accurate rainfall estimation is proposed based on these assumptions. The Bayesian analysis has the advantage of a statistically post-processing idea that could yield a predictive distribution with quantitative uncertainty (Renard, 2011; Shrestha et al., 2015). For example, a Bayesian kriging approach, which assumes a Gaussian process of precipitation at any location and considers the elevation a covariate, is developed for merging monthly satellite and gauge precipitation data (Verdin et al., 2015). A dynamic Bayesian model averaging (BMA) method, which shows better skill scores than the existing One-outlier removed (OOR) method, is applied for satellite precipitation data fusion across the TP (Ma et al., 2018; Shen et al., 2014). Given the challenges of quantifying precipitation biases in regions with complex terrain (Derin et al., 2019), continuous efforts are required to exert the potential merit of Bayesian analysis on this critical issue.

In this study, a two-stage blending (TSB) approach is proposed for merging multiple satellite precipitation estimates (SPE) and ground observations. The experiment is performed in the warm season (from May to September) during 2010-2014 in the northeastern TP (NETP), where a relatively denser network of rain gauges is available compared to other regions of TP. The TSB method is expected to help with the exploration of multi-source/scale precipitation data fusion in regions with complex terrain.

The remainder of this paper is organized below: Section 2 describes the experiment including the study region and precipitation data. Section 3 details the methodology, including the TSB approach, and two existing fusion methods (i.e., BMA and OOR). Results and discussions are presented in Sections 4 and 5, respectively. The primary findings are summarized in Section 6.

**2 Study area and data**

The study domain is located in the upper Yellow River basin of NETP (Fig. 1). As shown in the 90-m digital elevation data, the altitude ranges from 785 m in the northeast to 6252 m in the southeast. The total annual precipitation is around 500 mm and the annual mean temperature is 0.7°C (Cuo et al., 2013). To avoid snowfall contamination on the gauge observation in the cold season, satellite and ground precipitation data from the warm season (May to September) of 2010 to 2014 are collected for the case study.

Four mainstream SPE are used, including Precipitation Estimation from Remotely Sensed Information using Artificial Neural Networks - Climate Data Records (PERCDR) (Ashouri et al., 2015), Tropical Rainfall Measuring Mission (TRMM) Multi-satellite Precipitation Analysis (TMPA) 3B42 version 7 (3B42V7) (Huffman et al., 2007), National Oceanic and Atmospheric Administration (NOAA) Climate Prediction Centre (CPC) Morphing Technique Global Precipitation Analyses Version 1 (CMORPH) (Xie et al., 2017), and the Integrated Multi-satellitE Retrievals for the Global Precipitation Measurement (GPM) mission V06 Level 3 final run product (IMERG) (Huffman et al., 2018). The basic information of SPE is shown in Table 1. The IMERG has a 0.10° x 0.10° resolution, and other SPE have a spatial resolution of 0.25° x 0.25°. To eliminate the scale difference in the fusion process, IMERG is resampled from 0.10° to 0.25° using the nearest neighbour interpolation method in advance.

The China Gauge-based Daily Precipitation Analysis (CGDPA) is used as ground precipitation source. It is developed based on a rain gauge network of 2400 gauge stations in Mainland China using a climatology-based optimal interpolation and topographic correction algorithm (Shen and Xiong, 2014). The 34 grid cells with the gauge sites in the regions of interest are

assumed as ground references (GR), and all of the grid cells are independent from the Global Precipitation Climatology Center (GPCC) stations, which are used for bias correction of the TRMM/GPM-era data (e.g., 3B42V7 and IMERG), and CMORPH (Huffman et al., 2007; Hou et al., 2014; Xie et al., 2017).

## 3 Methodology

### 3.1 TSB

The diagram of the TSB method is shown in Figure 2. Stage 1 is designed to reduce the bias of the original SPE based on the GR at the training sites with a Bayesian correction (BC) procedure. In Stage 2, a Bayesian weighting (WS) model is used to merge the bias-corrected SPE.

#### 3.1.1 Bias correction

(a) Model structure

Let $R(s,t)$ denote near-surface precipitation at the GR cell $s$ and the $t^{th}$ day. The original SPE and bias-corrected SPE of PERCDR, 3B42V7, CMORPH and IMERG at the GR cell $s$ and the $t^{th}$ day are defined as $(Y_1(s,t), Y_2(s,t), ..., Y_p(s,t)$ and $(Y_1'(s,t), Y_2'(s,t), ..., Y_p'(s,t))$, respectively. For simplification purpose and without losing generality, these data at a particular GR cell and day will be denoted by $R$, $(Y_1, Y_2, Y_3, Y_4)$, and $(Y_1', Y_2', Y_3', Y_4')$. While for all GR cells and days, they will be denoted

in bold $\boldsymbol{R}$, $(\boldsymbol{Y_1, Y_2, Y_3, Y_4})$, and $(\boldsymbol{Y_1', Y_2', Y_3', Y_4'})$.

In Stage 1, we perform a conditional modelling of GR on each SPE, i.e., the probabilistic distribution $f(R)$ to improve the accuracy of the original SPE. Given that an appropriate assumption of $f(R)$ is necessary, the goodness-of-fit of the Lognormal, Gaussian, and Gamma distribution for the GR is examined graphically by using a probability-probability (PP) plot at the

training sets (Fig. 3). It is found that the usage of a Gamma distribution is more reliable as the associated PP plot is closer to the diagonal line than the others. For each satellite product, the Gamma distribution is parameterized as follows:

$$R \sim Gamma\left(\alpha_i, \frac{\alpha_i}{\mu_i}\right) \tag{1}$$

where $i$ is the number of satellite product. $\alpha_i$, $\mu_i$ and $\frac{\alpha_i}{\mu_i}$ are the shape, mean and rate parameters of the Gamma distribution, respectively. Let the $i^{th}$ SPE $Y_i$ and the associated terrain feature $Z$ be covariates related to the GR, the mean $\mu_i$ in Eq. (1) can

be described with generalized linear regression of covariates $Y_i$ and $Z$, which is written as follows:

$$\log(\mu_i) = \delta_i + \beta_i * \log(Y_i) + \gamma_i * Z \tag{2}$$

where $Z$, ranging from 0 to 1, is the normalized elevation feature of each site. $\boldsymbol{\theta_i} = \{\alpha_i, \delta_i, \beta_i, \gamma_i\}$ $(i = 1, ..., 4)$ is summarized as a parameter set and will be estimated in Bayesian framework. In the following, $\boldsymbol{Z}$ will be denoted as the collection of the normalized elevation feature for all training data.


According to the Bayes' theorem, the posterior probability density function (PDF) of parameter set $\boldsymbol{\theta_i}$ is expressed as:

$$f(\boldsymbol{\theta_i}|\mathbf{R}, \boldsymbol{Y_i}, \boldsymbol{Z}) \propto f(\boldsymbol{R}|\boldsymbol{\theta_i}, \boldsymbol{Y_i}, \boldsymbol{Z}) f(\boldsymbol{\theta_i}) \tag{3}$$

where $f(\boldsymbol{\theta_i})$ is the prior distribution and implies parameter information other than GR and SPE data, and $f(\boldsymbol{R}|\boldsymbol{\theta_i}, \boldsymbol{Y_i}, \boldsymbol{Z})$ is the likelihood function that defines the conditional probability of GR on the SPE and elevation. The priors of $f(\boldsymbol{\theta_i})$ are initialized as Cauchy distribution with $\alpha_i$ in terms of its location at zero and scale as $\sigma_{\alpha_i}$ in Eq. (4), and Gaussian distribution with $\delta_i, \beta_i, \gamma_i$ in terms of its mean at zero and standard deviation (SD) at $\sigma_{\delta_i}$ $\sigma_{\beta_i}$ $\sigma_{\gamma_i}$ in Eqs. (5) - (7), respectively.

$$\alpha_i \sim Cauchy(0, \sigma_{\alpha_i}) \tag{4}$$

$$\delta_i \sim Normal(0, \sigma_{\delta_i}) \tag{5}$$

$$\beta_i \sim Normal(0, \sigma_{\beta_i}) \tag{6}$$

$$\gamma_i \sim Normal(0, \sigma_{\gamma_i}) \tag{7}$$


Given that the assumption of the weakly informative priors ensures the Bayesian inference in an appropriate range (Ma et al., 2020b), the hyper-priors of $\sigma_{\alpha_i}, \sigma_{\delta_i}, \sigma_{\beta_i}, \sigma_{\gamma_i}$ are specified as 2, 10, 10, 10, respectively.

(b) Parameter estimation

The estimation of the posterior distribution $f(\boldsymbol{\theta_i}|\mathbf{R}, \boldsymbol{Y_i}, \boldsymbol{Z})$ in Eq. (3) becomes difficult as its dimension grows with the number of parameters (Renard, 2011; Ma and Chandrasekar, 2020). Robertson et al. (2013) obtained the maximum a posteriori (MAP) solution for model parameters using a stepwise method. Here, the Markov Chain Monte Carlo (MCMC) technique with its sampling algorithm as the No-U-Turn Sampler (NUTS) variant of Hamiltonian Monte Carlo in the Stan program is performed to address this issue (Gelman et al., 2013). The sampling records of model parameters are obtained based on the training data
in warm season of 2014 in the NETP. Since we only have four parameters in this model, the MCMC converges very quickly.

Thus, we run a chain of length 2000, removing the first 1000 iterations as the warm-up period and retaining the second 1000 iterations. The parameter samples of these 1000 iterations are the samples of the posterior distribution $f(\boldsymbol{\theta}_i|\boldsymbol{R}, \boldsymbol{Y}_i, \boldsymbol{Z})$.

(c) Bayesian inference

Based on the posterior distribution of parameter set $\boldsymbol{\theta}_i$ of each SPE, calculating the bias-corrected SPE $R^*$ at new site is of interest. It can be quantitatively simulated from its posterior distribution in Eq. (8) using the associated SPE $Y_i^*$, normalized elevation $Z_i^*$ and training data $\boldsymbol{R}, \boldsymbol{Y}_i, \boldsymbol{Z}$:

$$f(R^*|Y_i^*, Z_i^*, \boldsymbol{R}, \boldsymbol{Y}_i, \boldsymbol{Z}) = \int f(R^*, \boldsymbol{\theta}_i|Y_i^*, Z_i^*, \boldsymbol{R}, \boldsymbol{Y}_i, \boldsymbol{Z}) \, d\boldsymbol{\theta}_i \tag{8}$$

Following the rule of joint probabilistic distributions, the right term inside the integral of Eq. (8) can be written as:

$$f(R^*, \boldsymbol{\theta}_i|Y_i^*, Z_i^*, \boldsymbol{R}, \boldsymbol{Y}_i, \boldsymbol{Z}) = f(R^*|Y_i^*, Z_i^*, \boldsymbol{R}, \boldsymbol{Y}_i, \boldsymbol{Z}, \boldsymbol{\theta}_i)f(\boldsymbol{\theta}_i|Y_i^*, Z_i^*, \boldsymbol{R}, \boldsymbol{Y}_i, \boldsymbol{Z}) \tag{9}$$

Given that the new bias-corrected SPE $R^*$ is independent to the training data, the first term of the right side in Eq. (9) is transformed as:

$$f(R^*|Y_i^*, Z_i^*, \boldsymbol{R}, \boldsymbol{Y}_i, \boldsymbol{Z}, \boldsymbol{\theta}_i) = f(R^*|Y_i^*, Z_i^*, \boldsymbol{\theta}_i) \tag{10}$$

Since the parameters $\boldsymbol{\theta}_i$ are only dependent upon the training data $\boldsymbol{R}, \boldsymbol{Y}_i, \boldsymbol{Z},$ the second term of the right side in Eq. (9) is expressed as:

$$f(\boldsymbol{\theta}_i|Y_i^*, Z_i^*, \boldsymbol{R}, \boldsymbol{Y}_i, \boldsymbol{Z}) = f(\boldsymbol{\theta}_i|\boldsymbol{R}, \boldsymbol{Y}_i, \boldsymbol{Z}) \tag{11}$$

Therefore, the predictive PDF of $R^*$ in Eq. (8) is written below:

$$f(R^*|Y_i^*, Z_i^*, \boldsymbol{R}, \boldsymbol{Y}_i, \boldsymbol{Z}) = \int f(R^*|Y_i^*, Z_i^*, \boldsymbol{\theta}_i)f(\boldsymbol{\theta}_i|\boldsymbol{R}, \boldsymbol{Y}_i, \boldsymbol{Z}) \, d\boldsymbol{\theta}_i \tag{12}$$

Since there is no general way to calculate the associated integral in Eq. (12), the prediction is performed using the MCMC iterated samplings (Renard, 2011). As for each SPE, a numerical algorithm is suggested below, where $n_{sim}$ stands for the replicate of the post-convergence MCMC samples and is set as 1000 in the case study. Thus, the predicted samples for $R^*$ in Eq. (12) are iterated ($k = 1:n_{sim}$) as follows:

1) For the $i^{th}$ satellite product, randomly select a parameter sample $\boldsymbol{\theta_i}=\{\alpha_i, \delta_i, \beta_i, \gamma_i\}$ from the MCMC samples;

2) Generate a value $R_k^*$ from a $Gamma\left(\alpha_i, \frac{\alpha_i}{\mu_i^*}\right)$, where $\log(\mu_i^*) = \delta_i + \beta_i * Y_i^* + \gamma_i * Z^*$;

Repeating step 1 and 2 for $n_{sim}$ times, the samples $R_k^*$ ($k = 1:n_{sim}$) are regarded as the realizations of the distribution of the bias-corrected SPE associated to the satellite estimation $Y_i^*$ and normalized elevation $Z^*$. The mean value of the samples $R_k^*$, denoted by $Y_i'$, is regarded as the bias-corrected SPE and the associated credible intervals (e.g., 2.5% and 97.5% quantiles) is used for predictive uncertainty.

### 3.1.2 Data merging

Ideally, the blended SPE ($B$) should be close to GR, i.e., $R$. Given the Gamma distribution of GR in Step 1, the blended SPE can be parameterized below:

$$B \sim Gamma\left(\alpha_B, \frac{\alpha_B}{\mu_B}\right) \tag{13}$$

where $\alpha_B, \mu_B$ and $\frac{\alpha_B}{\mu_B}$ are the shape, mean and rate parameters, respectively. In this step, the bias-corrected SPE of 4 satellites are merged with weight parameters $w_i (i = 1, \ldots, 4)$, and $\varepsilon$ is the residual error. The data fusion of bias-corrected SPE specified in the *log* scale is defined as follows:

$$\log(\mu_B) = \sum_{i=1}^{4} \log(Y_i') * w_i + \varepsilon \tag{14}$$

$$\sum_{i=1}^{4} w_i = 1 \tag{15}$$

$$\varepsilon \sim Normal(0, \sigma_\varepsilon) \tag{16}$$

Thereby, all parameters including $\alpha_B, w_i (i = 1,2,..4)$ and $\sigma_\varepsilon$ can be estimated from the GR and bias-corrected SPE at the training sites. The estimation process in a Bayesian framework is similar to that described in Stage 1. After all parameters are estimated, as similar to the Bayesian inference in Stage 1, the blended SPE at any site and time can be derived with the bias-corrected SPE and corresponding weights using the MCMC iterations.

### 3.2 Comparison model

#### 3.2.1 BMA

The BMA method is a statistical algorithm that merges predictive ensembles based on the individual SPE at the training period in regions of interest. Here, the BMA result refers to the ensemble SPE. Based on the law of total probability, the conditional probability of the BMA data on the individual SPE is expressed as:

$$f(BMA|Y_1, \ldots, Y_p) = \sum_{i=1}^{p} f(BMA|Y_i) \cdot w_i \qquad (17)$$

where $f(BMA|Y_i)$ is the predictive PDF given by the individual SPE $Y_i$ and $w_i$ is the corresponding weight. The log-likelihood function $l$ is applied to calculate the BMA parameter set $\boldsymbol{\vartheta}$, which is written as:

$$l(\boldsymbol{\vartheta}) = \log\left(\sum_{i=1}^{p} w_i \times f(BMA|Y_i)\right) \qquad (18)$$

It is assumed that $f(BMA|Y_i)$ follows a Gaussian distribution with its parameters as $\theta_i$, and BMA is ideally close to GR at any site and time. Eq. (18) is written as:

$$l(\boldsymbol{\vartheta}) = \log(\sum_{i=1}^{p} w_i \times g(GR|\theta_i)) \qquad (19)$$

where $g(\cdot)$ stands for Gaussian distribution, and $\boldsymbol{\vartheta} = \{w_i, \theta_i, i = 1, \ldots, p\}$. The optimal BMA parameters $\boldsymbol{\vartheta}$ are calculated by maximizing the log likelihood function using the expectation–maximization algorithm. Before executing the BMA method, both GR and SPE data are pre-processed using the Box-Cox transformation to ensure that $f(BMA|Y_i)$ $(i = 1, \ldots, 4)$ is close to Gaussian distribution. As the BMA weights, $w_i, i = 1, \ldots, 4$ are obtained, the BMA data is calculated by weighted sum of the original SPE at any site and time. More details of the BMA method can be found in Ma et al. (2018).

#### 3.2.2 OOR

The OOR method is defined as the arithmetic mean of the individual SPE by removing the feature with the largest offset. It is written as:

$$OOR = \frac{1}{p-1}\sum_{i=1}^{p-1} Y_i \qquad (20)$$

where $Y_i$ is the individual SPE, $p$ is the number of SPE. The original SPE with the largest offset among the satellite products is removed and the average of the remaining SPE is regarded as the OOR result.

## 3.3 Error analysis

To assess the performance of the proposed TSB method, several statistical error indices including root mean square errors (RMSE), normalized mean absolute errors (NMAE), and the Pearson's correlation coefficients (CC) are used in this study. The specific formulas of these metrics can be found below:

$$RMSE = \sqrt{< (Sim - Obs)^2 >} \tag{21}$$

$$NMAE = \frac{<|Sim - Obs|>}{<Obs>} \times 100\% \tag{22}$$

$$CC = \frac{\sum[(Sim - <Sim>)(Obs - <Obs>)]}{\sqrt{\sum(Sim - <Sim>)^2}\sqrt{\sum(Obs - <Obs>)^2}} \tag{23}$$

where *Sim* and *Obs* stand for the simulated and observed data, respectively; the angle brackets stand for sample average.

## 4 Results

In the experiment, model parameters are calibrated on the daily precipitation of warm season in 2014, where GR data at the 27 black grids in Figure 1 are randomly selected for training the model. The model validation is performed under two scenarios: Scenario 1 will validate the model in space based on the data of the same period in validation stations (i.e., the 7 red grids in Figure 1), and Scenario 2 will validate the model in time based on the data of warm season from 2010 to 2013 at the same 27 black grids in Figure 1. In addition, we consider a 10-fold cross validation in space by randomly selecting 7 sites for model validation, and the data of the remaining 27 sites as the training set. The performance of TSB approach is further compared with BMA and OOR in the two scenarios.

## 4.1 Parameter estimates

Figures 4 and 5 show the posterior distribution curves of the posterior parameters in Stage 1 and 2, respectively. As for each parameter in the bias-corrected process, the individual SPE including PERCDR, 3B42V7, CMORPH and IMERG shows similar pattern (Figs. 4a to 4d). It shows that the bias structures of the original SPE have similar characteristics. For all SPE, the distribution mass of parameter $\beta_i$ are all on the right side of zero, which implies that a systematic bias exists for all satellite products. When looking at the effects of elevation, the posterior distribution of parameter $\gamma_i$ for PERCDR, 3B42V7 and CMORPH (Figs. 4a, 4b and 4c) have value zero in the middle range of the distribution, which implies that elevation may have little impacts on these three satellite products. While for IMERG in Fig. 4d, the distribution mass of parameter $\gamma_i$ is mostly on the right side of zero, which implies a clear effect of elevation on this satellite product. In the data fusion step (Fig. 5), IMERG has the highest weight and PERCDR has the lowest weight among the four bias-corrected SPE. Moreover, 3B42V7 and

PERCDR have similar contribution on the blended result. Basically, Bayesian analysis is able to simulate the parameter uncertainty as compared with the traditionally statistical methods.

## 4.2 Model validation under two scenarios

Table 2 presents the summary of the statistical error indices including RMSE, NMAE and CC of the original (i.e., PERCDR, 3B42V7, CMORPH and IMERG), bias-corrected (i.e., BC-PER, BC-V7, BC-CMO and BC-IME) and blended SPE under two scenarios in the NETP. The sub-section 4.2.1 and 4.2.2 show the performance of the model validation under Scenario 1 and 2, respectively.

### 4.2.1 Scenario 1

The original SPE show large biases with the RMSE, NMAE, and CC indices ranging from 6.25-8.56 mm/d, 60.6-80.3%, and 0.382-0.556, respectively. 3B42V7 has the worst skill with the highest RMSE of 8.56 mm/d, the highest NMAE of 80.3% and the second lowest CC of 0.383. CMORPH shows the best performance with the lowest RMSE of 6.25 mm/d, the lowest NMAE of 60.6% and the highest CC of 0.556, which presents its superiority compared with the other original SPE in the NETP. Based on the BC model, all the bias-corrected SPE have better agreements with GR compared with the original SPE. Their RMSE 250   scores range from 5.43 to 6 mm/d, and decrease by 13~31.8%, and their NMAE scores vary from 56.0 to 63.5%, and decline by 7.1 to 23.5%, respectively. Meanwhile, their CC values range from 0.346 to 0.533 after bias correction. With the BW model, the blended SPE is closer to GR in terms of RMSE, NMAE and CC at 5.36 mm/h, 54.6%, and 0.57, respectively, compared with both the original and bias-corrected SPE. The RMSE and NMAE values of the blended SPE decrease by 14.3~37.4% and 10~32%, respectively, and the CC value increases by 2.4~49.2%, accordingly, compared to the original SPE. In addition, the 255   RMSE, NMAE and CC of the blended SPE increases by 1.4~10.8%, 2.5~14.1%, and 6.8~64.8%, respectively, compared with the bias-corrected SPE. It proves that the blended SPE exhibits higher quality after Stage 2, due to the ensemble contribution of the bias-corrected SPE. The relative weight of BC-PER, BC-V7, BC-CMO and BC-IME is 0.02, 0.038, 0.295, and 0.647, respectively. The BC-IME and BC-PER have the highest and lowest weights, respectively, and the BC-V7 and BC-CMO rank between BC-IME and BC-PER (Fig. 6a). As for the original SPE, it is found that there is an overestimation when the rainfall 260   is less than 7.6 mm/d, and an underestimation when the rainfall is more than 7.6 mm/d. Based on the proposed TSB approach, the blended SPE is closer towards the GR (Figs. 6b and 6c). Meanwhile, BC-PER seems to be clearly different from the other bias-corrected SPE, and to this point in the study has shown little value to be kept in consideration in the merging process. However, it is worth noting that PERCDR can in fact be informative and on a case by case basis.

### 4.2.2 Scenario 2

The proposed TSB approach is also validated in Scenario 2, where the blended SPE shows better performance in terms of its RMSE, NMAE and CC at 6.37 mm/h, 56.7% and 0.513, respectively, compared with both the original and bias-corrected SPE. It shows that the original SPE including PERCDR, 3B42V7, CMORPH and IMERG have high RMSE and NMAE scores in terms of 7.20~9.19 mm/h and 61.9~79.3%, respectively, and low CC values in terms of 0.261~0.493. After the bias correction, the four satellite products have increased their performance with lower error indices than the original SPE. The RMSE indices

of the bias-corrected SPE vary from 6.41 to 7.03 mm/h, and the corresponding NMAE and CC indices are from 57.7% to 64.5%, and from 0.253 to 0.48, respectively. Based on the data fusion process, the error indices of the blended SPE including RMSE, NMAE and CC are 6.37 mm/h, 56.7% and 0.513, respectively. It is found that the RMSE and NMAE values of the blended SPE decreased by 11.5~30.7% and 8.4~28.5%, respectively, and the CC value increases by 4.1~96.6% compared with the original SPE.

As learned from the two validated scenarios, it proves that the TSB approach has the potential in improving the satellite rainfall accuracy, and it has its ability to exert benefits from SPE in terms of higher performances and mitigate poor impacts from the ones with lower quality.

### 4.3 Cross-validation

Figures 7 and 8 show the statistics of evaluation scores of RMSE, NMAE, and CC for the original SPE and blended estimates at the validation grids with 10 random split of the gauge locations in the warm season of 2014. For each test, 7 grid sites are randomly selected from the 34 grid cells and used for model verification, and the remaining 27 grid sites are used for training the model.

As for the blended SPE, it performs similar scores at the validation grids among the 10-fold random samples. The blended SPE shows better skill compared with the original SPE at each test in terms of RMSE, NMAE, and CC (Fig. 7). Statistically, the mean values of RMSE, NMAE and CC for the blended SPE are 5.75 mm/h, 57.1% and 0.551, respectively (Table 3). The averaged improvement ratios of RMSE for the blended SPE are 27.6%, 25%, 10.6% and 13% compared to the PERCDR, 3B42V7, CMORPH and IMERG, respectively, and similar performance is seen from NMAE with the average improvement

ratios of 24.5%, 22.3%, 7.8% and 7.3%, respectively (Table 4). In summary, the 10-fold cross validation further verified that the blended SPE has a higher accuracy of gridded precipitation than the original satellite products.

## 4.4 Model comparison with BMA and OOR

To assess the performance of the proposed TSB approach, it is beneficial to compare the TSB result with the existing fusion approach. In this study, the BMA approach makes use of four original satellite data and the corresponding GR data at the 27 black grids shown in Figure 1 in the warm season of 2014 to estimate the optimal BMA weights. In Scenario 1, the BMA data are calculated based on the BMA weights and the original SPE from the 7 red grids in the warm season of 2014, and the OOR data are calculated based on the OOR method using the original SPE data from the 7 red grids in the warm season of 2014. In Scenario 2, the BMA data are calculated based on the BMA weights and the original SPE from the 27 black grids in the warm season from 2010 to 2013, and the OOR result are calculated based on the OOR method and the original SPE data from the 27 black grids in the warm season from 2010 to 2013. Herein, we compare the blended SPE with both of the BMA and OOR predictions in two scenarios and their statistical error summary is shown in Table 5.

In Scenario 1, the TSB method performs better skill scores with the RMSE, NMAE and CC values of 5.36 mm/d, 54.6%, and 0.57, respectively, as compare with the BMA and OOR approaches. In addition, OOR shows the worst performance in terms of RMSE, NMAE, and CC at 6.22 mm/d, 59.7%, and 0.537, respectively. BMA shows better skill than OOR but worse skill than TSB, in terms of the RMSE, NMAE and CC values at 5.78 mm/d, 56.6% and 0.562, respectively. In Scenario 2, similar performance is found for the TSB approach, where it has lower RMSE (6.37 mm/d) and NMAE (56.7%) and higher CC (0.513) than both the OOR and BMA results. Basically, as compared with the two existing fusion algorithms (BMA and OOR) in the two validated scenarios, it confirms that the TSB method has an advantage for combining the original SPE and reducing the bias of the satellite precipitation retrievals. It is noted that the daily precipitation estimates follow a gamma distribution (Eq. (1)) in this study, in future work it would be interesting to examine whether the gamma distribution can be used in the BMA algorithm without converting it to a Gaussian distribution.

## 4.5 Model performance on a heavy rainfall case

Local recycling plays as a premier role for the moisture sources of rainfall extremes in the NETP (Ma et al., 2020a). The September 22, 2014 event is a storm that would represent the local heavy rainfall pattern in the warm season. Considering that accurate precipitation estimate on extreme weather is very important for flood hazard mitigation, we investigate the utility of the proposed TSB approach on this event to quantify its performance in extreme rainfall case (Fig. 9a). The relative weights of BC-PER, BC-V7, BC-CMO, and BC-IME for the blended SPE are 0.264, 0.14, 0.191 and 0.405, respectively, on this event (Fig. 9b). It is found that the IMERG data has the biggest contribution and the 3B42V7 and CMORPH data have nearly similar contribution for the blended SPE.

Table 6 reports the evaluation statistics reflecting the blended performance on this case. It shows that the RMSE, NMAE and CC values of the original SPE range from 8.18~9.24 mm/d, 47~52.8%, and 0.642~0.85, respectively. Compared to the original SPE, the blended SPE has lower RMSE of 5.23 mm/d, and lower NMAE of 31.5%, and higher CC of 0.837, respectively. The RMSE and NMAE values of the blended SPE decrease by 36.1~43.4% and 33~40.3%, respectively. The performance of the TSB approach is further explored at three gauge cells (i.e., IDs 56171, 56173, 56067) with the top three daily rainfall records on September 22, 2014. Figure 10 shows the PDF curves of blended samples at the three sites in this case. It demonstrates that the blended SPE has the advantage in quantifying the predictive uncertainty on rainfall extremes at each site. For example, at ID 56171, the rainfall estimates that are derived from the original SPE are 19.8 mm (PERCDR), 35.3 mm (3B42V7), 26 mm (CMORPH), and 21.2 mm (IMERG), respectively. 3B42V7 shows an overestimation, while PERCDR, CMORPH and IMERG underperform the daily rainfall at the corresponding pixel (Fig. 10a). Based on the proposed TSB approach, the mean value of the merging estimates are 28 mm/d. At IDs 56173 and 56067, the mean values of the blended SPE are 26.2 and 19.7 mm/d, respectively, and they are close to GR with the daily amounts at 30.9 and 28.7 mm, respectively (Figs. 10b and 10c). Overall, these analyses reveal that the TSB algorithm could not only quantify its predictive uncertainty, but also improve the daily rainfall amount even under heavy rainfall conditions.

## 4.6 Model application in a spatial domain

It is important to explore the Bayesian ensembles at unknown sites in the domain. As learned from Figure 11, it seems that each of the original SPE can capture the spatial pattern of daily mean precipitation in the warm season, but might fail in the representation of precipitation amount, partly because of the satellite retrieval bias in complex terrain and limited GR network. Thus, the TSB method is further applied in the region of interest to demonstrate its performance on daily precipitation in the warm season of 2010-2014 in the NETP. It is found that the blended SPE shows high precipitation in the southwest and low precipitation in the northwest, as well as moderate precipitation in the eastern region. In addition, as compared with the original SPE, higher values disappear from the spatial map except in southwest corner for the blended SPE. The possible reason is that daily mean rainfall is the highest in southwest corner for most SPE, and larger value exists after the TSB approach. Meanwhile, the predictive Bayesian uncertainties including lower (2.5%) and upper (97.5%) quantiles are displayed from Figures. 12b to 12c to illustrate the blending variation in this application.

**5 Discussion**

In spite of the superior performance of the TSB algorithm, some issues still need to be considered in the practical applications:

Because of limited knowledge on the influences of complex terrain and local climate on the rainfall patterns in the study area, the elevation feature is considered in the first stage. Table 7 quantifies the impact of elevation covariate on the bias-corrected and blended SPE performances in Scenario 1 in the warm season of 2014 in the NETP. It is found that the inclusion of elevation feature provides slightly better skill compared with the results without terrain information in this experiment. Considering that deep convective systems occurring near the mountainous area have an effect on the precipitation cloud (Houze, 2012), more attempts are required to improve the orographic precipitation in the TP in future.

The data fusion application is based on four mainstream SPE, and BC-IME and BC-PER show the best and worst performances among the bias-corrected SPE in Stage 1. It raises a question that why not simply apply the first stage of bias correction and then select the best-performed bias-corrected SPE as the final product. To address this issue, we investigate the statistical error differences among the BC-IME and blended SPE before and after the removing of BC-PER for 10-fold cross validation in the warm season of 2014 in the NETP (Fig. 13). It is beneficial to involve the Stage 2 in the TSB method because the blended SPE performs better skill than the best-performed bias-corrected SPE (i.e., BC-IME) in Stage 1. The primary reason is that the BW model is designed to integrate various types of bias-corrected SPE, which is limited in the BC model. In addition, both the blended SPE with and without the consideration of PERCDR show similar performances of the RMSE, NMAE, and CC indices (Fig. 13). It implies that the TSB approach has an advantage of not impacted by the poor quality individuals (e.g., BC-PER), partly because the BW model can reallocate the contribution of the bias-corrected SPE based on their corresponding bias characteristics.

In addition, as calculating the blended result at any new sites, the model parameters derived from the training grid sites are assumed to be applicable in the whole domain. Since we have a relatively dense GR network in the survey region, the current assumption is acceptable according to the performance of the blended SPE. It is helpful to give some guideline on how many training sites are needed to apply the TSB approach in a region with complex terrain and limited GR. The sensitivity analysis of the number of training grid cells on the performance of blended SPE at the validation grids is explored in Figure 14. As the number of training sites is increasing, there is a decreasing trend for the RMSE and NMAE values, but a slight increasing trend for the CC value. It seems that the performance of the blended SPE becomes similar as the number of training sites increases to 21. We admit that the more information from the ground observations, it would be more beneficial for the blended gridded

product in the region of interest. It is noted that, if extended to the TP or global scale, the extension of model parameters and training sites should be carefully considered. For instance, there are few gauges installed in the western and central TP (Ma et al., 2015), it might be a potential risk to directly apply this fusion algorithm for these regions.

The aim of this study is not to model rainfall process in a target domain, but to propose an idea to extract valuable information from available SPE and provide more reliable gridded precipitation in high-cold region with complex terrain. Considering its spatiotemporal differences and the existence of many zero-value records, rainfall is extremely difficult to observe and predict (Yong et al., 2015; Bartsotas et al., 2018). With regard to the probability of rainfall occurrence, a zero-inflated model, which is coherent with the empirical distribution of rainfall amount, is expected to improve the proposed TSB algorithm. Also, hourly
or even instantaneous precipitation intensity is extremely vital for flood prediction, which should be specifically designed when extending this fusion framework in the next step.

## 6 Summary and prospects

This study proposes a TSB algorithm for multi-SPE data fusion. A preliminary experiment is conducted in the NETP using four mainstream SPE (i.e., PERCDR, 3B42V7, CMORPH, and IMERG) to demonstrate the performance of this TSB approach.
Primary conclusions are summarized below:

(1) This TSB algorithm has two stages and involves the BC and BW models. It is found that this blended method is capable of involving a group of original SPE. Meanwhile, it provides a convenient way to quantify the fusion performance and the associated uncertainty.


(2) The experiment shows that the blended SPE has better skill scores compared to the original SPE in the two validated scenarios. The 10-fold cross validation in Scenario 1 further confirms the superiority of the TSB algorithm. In addition, it is found that the TSB method outperforms another two existing fusion methods (i.e., BMA and OOR) in the two scenarios. The performance of this fusion method is also demonstrated under a heavy rainfall event in the region of interest.


(3) The application proves that this algorithm can allocate the contribution of individual SPE on the blended result because it is capable of ingesting useful information from uneven individuals and alleviating potential negative impacts from the poorly performing members.

Overall, this work provides an opportunity for merging SPE in high-cold region with complex terrain. The evaluation analysis of this TSB method for extended regions (e.g., TP) in terms of higher temporal resolution (e.g., hourly) will be performed in a future study.

*Data availability*

The gauge data are from China Meteorological Data Service Centre (http://data.cma.cn). The PERCDR data are obtained from
http://www.ncei.noaa.gov/data/precipitation-persiann/; the CMORPH data are from https://rda.ucar.edu/datasets/ds502.2/; the 3B42V7 data are from https://disc2.gesdisc.eosdis.nasa.gov/data/TRMM_L3/TRMM_3B42_Daily.7/; the IMERG data are from https://gpm1.gesdisc.eosdis.nasa.gov/data/GPM_L3/GPM_3IMERGDF.06/.

*Author contributions*

YM and XS conceived the idea; XS and YZ acquired the project and financial supports. YM conducted the detailed analysis;
HC, XS and YZ gave comments on the analysis; all the authors contributed to the writing and revisions.

*Competing interests*

The authors declare that they have no conflict of interest.

*Acknowledgements*

This study is supported by the National Key Research and Development Program of China (Nos. 2017YFC1503001 and
2017YFA0603101) and Strategic Priority Research Program (A) of CAS (No. XDA2006020102).

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

**Figure and Table Captions**

**Table 1:** Basic information of the original SPE used in this study.

**Table 2:** Summary of statistical error indices (i.e., RMSE, NMAE, and CC) of the original, bias-corrected, and blended SPE in two scenarios in the NETP.

**Table 3:** Summary of the mean values of RMSE, NMAE and CC for the original and blended SPE at 10 random verified tests in the warm season of 2014 in the NETP.

**Table 4:** Mean improvement ratios of statistical error indices of the blended SPE, in terms of RMSE, NMAE and CC compared with the original SPE at 10 random verified tests in the warm season of 2014 in the NETP.

**Table 5**: Summary of statistical error indices (i.e., RMSE, NMAE, and CC) for three fusion methods (i.e., OOR, BMA, and TSB) in the two scenarios in the NETP.

**Table 6:** Summary of statistical error indices (i.e., RMSE, NMAE, and CC) for the original and blended SPE during a heavy rainfall event of September 22, 2014 in the NETP.

**Table 7** Summary of statistical error indices (i.e., RMSE, NMAE, and CC) for bias-corrected and blended SPE with and without consideration of terrain feature as a covariate in the TSB method in Scenario 1 in the NETP.

**Figure 1:** Spatial map of the topography and GR network used in the study, where 27 black cells are used for model calibration and 7 red cells are for model verification.

**Figure 2:** The diagram of the proposed TSB algorithm.

**Figure 3:** (a) The histogram density plot and (b) the corresponding Probability-Probability plot of GR at the training grids in the warm season of 2014 in the NETP, where the red, blue and green lines shows the fitted Gamma, Lognormal and Gaussian distribution, respectively.

**Figure 4**: The PDF curves of posterior parameter sets with regard to (a) PERCDR, (b) 3B42V7, (c) CMORPH and (d) IMERG in the bias correction process, i.e., Stage 1.

**Figure 5**: The PDF curves of posterior parameter sets in the data fusion process, i.e., Stage 2.

**Figure 6:** (a) The Box-Whisker plots of relative weights for the bias-corrected SPE, (b) the scatter plots between GR and the original SPE and (c) the PDF of daily rainfall for the GR, original and blended SPE with various rain intensities in Scenario 1 in the NETP.

**Figure 7:** Statistical error indices of the original and blended SPE at 10 random verified tests in the warm season of 2014 in the NETP: (a) RMSE, (b) NMAE and (c) CC.

**Figure 8:** The Box-Whisker plots of improvement ratios of statistics for the blended SPE compared with the original SPE, including PERCDR, 3B42V7, CMORPH, and IMERG at 10 random verified tests in the warm season of 2014 in the NETP: (a) RMSE, (b) NMAE and (c) CC.

**Figure 9:** (a) Spatial pattern of gauge-based measurements during a heavy rainfall case of September 22, 2014 in the NETP, where the site IDs 56171, 56173 and 56067 report the top three daily rainfall amounts of 32.3 mm, 30.9 mm and 28.7 mm, respectively; (b) the corresponding Box-Whisker plots of relative weights of the bias-corrected SPE in the data fusion process.

**Figure 10:** The PDF curves of blended SPE samples and the corresponding mean value at three gauge-based grids on a heavy rainfall case of September 22, 2014: (a) ID 56171, (b) ID 56173 and (c) ID 56067. The original SPE and GR at each pixel are also indicated in each subfigure.

**Figure 11:** Spatial patterns of the daily mean precipitation in terms of the original SPE in the warm season of 2010 to 2014 in the NETP: (a) PERCDR, (b) 3B42V7, (c) CMORPH, and (d) IMERG.

**Figure 12:** Spatial patterns of the blended SPE in terms of (a) mean, (b) lower quantile (2.5%) and (c) upper quantile (97.5%) of daily mean precipitation in the warm season of 2010 to 2014 in the NETP.

**Figure 13.** Statistical error indices (i.e., RMSE, NMAE, and CC) of the best-performed bias-corrected SPE (i.e., BC-IME, black) and blended SPE before (red) and after (blue) removing the worst-performed BC-PER at 10 random verified tests in the warm season of 2014 in the NETP.

**Figure 14:** Statistical error indices (i.e., RMSE, NMAE, and CC) of the blended SPE at the validation grid locations in terms of different number of training sites in the warm season of 2014 in the NETP.

**Table 1:** Basic information of the original SPE used in this study.

| Short name | Full name and details | Temporal resolution | Spatial resolution | Input data | Retrieval algorithm | References |
|---|---|---|---|---|---|---|
| PERCDR | Precipitation Estimation from Remotely Sensed Information using Artificial Neural Networks (PERSIANN) Climate Data Record (CDR) | Daily | 0.25° x 0.25° | Warm season from 2010 to 2014 | Adaptive artificial neural network | *Ashouri et al.,* 2015 |
| 3B42V7 | TRMM Multi-satellite Precipitation Analysis (TMPA) 3B42 Version 7 | Daily | 0.25° x 0.25° | Warm season from 2010 to 2014 | GPCC monthly gauge observation to correct this bias of 3B42RT | *Huffman et al.,* 2007 |
| CMORPH | NOAA Climate Prediction Centre (CPC) Morphing Technique (CMORPH) Global Precipitation Estimates Version 1 | Daily | 0.25° x 0.25° | Warm season from 2010 to 2014 | Morphing technique | *Xie et al.,* 2017 |
| IMERG | Integrated Multi-satellitE Retrievals for the Global Precipitation Measurement (GPM) mission V06 Level 3 final run product | Daily | 0.10° x 0.10° | Warm season from 2010 to 2014 | 2017 version of the Goddard profiling algorithm | *Huffman et al., 2018* |

**Table 2:** Summary of statistical error indices (i.e., RMSE, NMAE, and CC) of the original, bias-corrected and blended SPE in two scenarios in the NETP.

| Scenarios | Product | RMSE (mm/d) | NMAE (%) | CC |
|---|---|---|---|---|
| Scenario 1 | PERCDR | 7.15 | 70.2 | 0.382 |
| | 3B42V7 | 8.56 | 80.3 | 0.383 |
| | CMORPH | 6.25 | 60.6 | 0.556 |
| | IMERG | 6.60 | 62.9 | 0.506 |
| | BC-PER | 6.00 | 63.5 | 0.346 |
| | BC-V7 | 5.83 | 61.4 | 0.408 |
| | BC-CMO | 5.43 | 56.3 | 0.533 |
| | BC-IME | 5.44 | 56.0 | 0.530 |
| | Blended SPE | 5.36 | 54.6 | 0.570 |
| Scenario 2 | PERCDR | 9.19 | 79.3 | 0.261 |
| | 3B42V7 | 8.38 | 71.3 | 0.403 |
| | CMORPH | 7.20 | 61.9 | 0.493 |
| | IMERG | 7.64 | 65.1 | 0.462 |
| | BC-PER | 7.03 | 64.5 | 0.253 |
| | BC-V7 | 6.69 | 61.3 | 0.395 |
| | BC-CMO | 6.41 | 58.2 | 0.480 |
| | BC-IME | 6.44 | 57.7 | 0.470 |
| | Blended SPE | 6.37 | 56.7 | 0.513 |

**Table 3:** Summary of the mean values of RMSE, NMAE and CC for the original and blended SPE at 10 random verified tests in the warm season of 2014 in the NETP.

| Product | RMSE (mm/d) | NMAE (%) | CC |
|---|---|---|---|
| PERCDR | 7.96 | 75.9 | 0.330 |
| 3B42V7 | 7.72 | 73.8 | 0.424 |
| CMORPH | 6.59 | 63.1 | 0.520 |
| IMERG | 6.78 | 62.7 | 0.518 |
| Blended SPE | 5.75 | 57.1 | 0.551 |

**Table 4:** Mean improvement ratios of statistical error indices of the blended SPE, in terms of RMSE, NMAE, and CC compared with the original SPE at 10 random verified tests in the warm season of 2014 in the NETP.

|  | Index | PERCDR | 3B42V7 | CMORPH | IMERG |
|---|---|---|---|---|---|
| Improvement Ratio (%) | RMSE (mm/d) | 27.6 | 25.0 | 10.6 | 13.0 |
|  | NMAE (%) | 24.5 | 22.3 | 7.8 | 7.3 |
|  | CC | 71.1 | 39.8 | 11.1 | 10.7 |

**Table 5**: Summary of statistical error indices (i.e., RMSE, NMAE, and CC) for three fusion methods (i.e., OOR, BMA, and TSB) in the two scenarios in the NETP.

| Scenarios | Method | RMSE (mm/d) | NMAE (%) | CC |
|---|---|---|---|---|
| | OOR | 6.22 | 59.7 | 0.537 |
| Scenario 1 | BMA | 5.78 | 56.6 | 0.562 |
| | TSB | 5.36 | 54.6 | 0.570 |
| | OOR | 7.04 | 59.9 | 0.498 |
| Scenario 2 | BMA | 6.79 | 58.8 | 0.500 |
| | TSB | 6.37 | 56.7 | 0.513 |

**Table 6:** Summary of statistical error indices (i.e., RMSE, NMAE, and CC) for the original and blended SPE during a heavy rainfall event of September 22, 2014 in the NETP.

| Product | RMSE (mm/d) | NMAE (%) | CC |
|---|---|---|---|
| PERCDR | 8.18 | 47.0 | 0.850 |
| 3B42V7 | 9.24 | 52.8 | 0.683 |
| CMORPH | 8.27 | 48.5 | 0.734 |
| IMERG | 8.63 | 49.1 | 0.642 |
| Blended SPE | 5.23 | 31.5 | 0.837 |

**Table 7** Summary of statistical error indices (i.e., RMSE, NMAE, and CC) for bias-corrected and blended SPE with and without consideration of terrain feature as a covariate in the TSB method in Scenario 1 in the NETP.

| Product | Type | RMSE (mm/d) | NMAE (%) | CC |
|---------|------|-------------|----------|-----|
| BC-PER | No Terrain | 5.98 | 63.3 | 0.361 |
| | Terrain | 6.00 | 63.5 | 0.346 |
| BC-V7 | No Terrain | 5.83 | 61.5 | 0.409 |
| | Terrain | 5.83 | 61.4 | 0.408 |
| BC-CMO | No Terrain | 5.48 | 56.9 | 0.520 |
| | Terrain | 5.43 | 56.3 | 0.533 |
| BC-IME | No Terrain | 5.48 | 56.3 | 0.519 |
| | Terrain | 5.44 | 56.0 | 0.530 |
| Blended SPE | No Terrain | 5.41 | 55.0 | 0.557 |
| | Terrain | 5.36 | 54.6 | 0.570 |


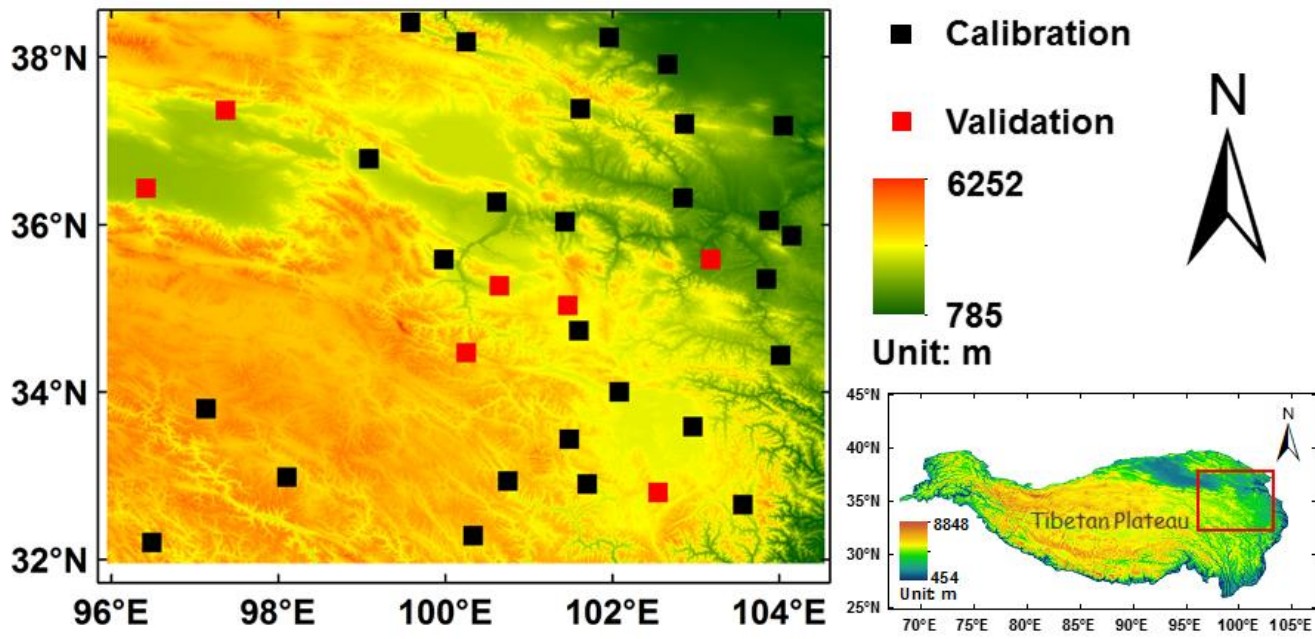

**Figure 1:** Spatial map of the topography and GR network used in the study, where 27 black cells are used for model calibration and 7 red cells are for model verification.


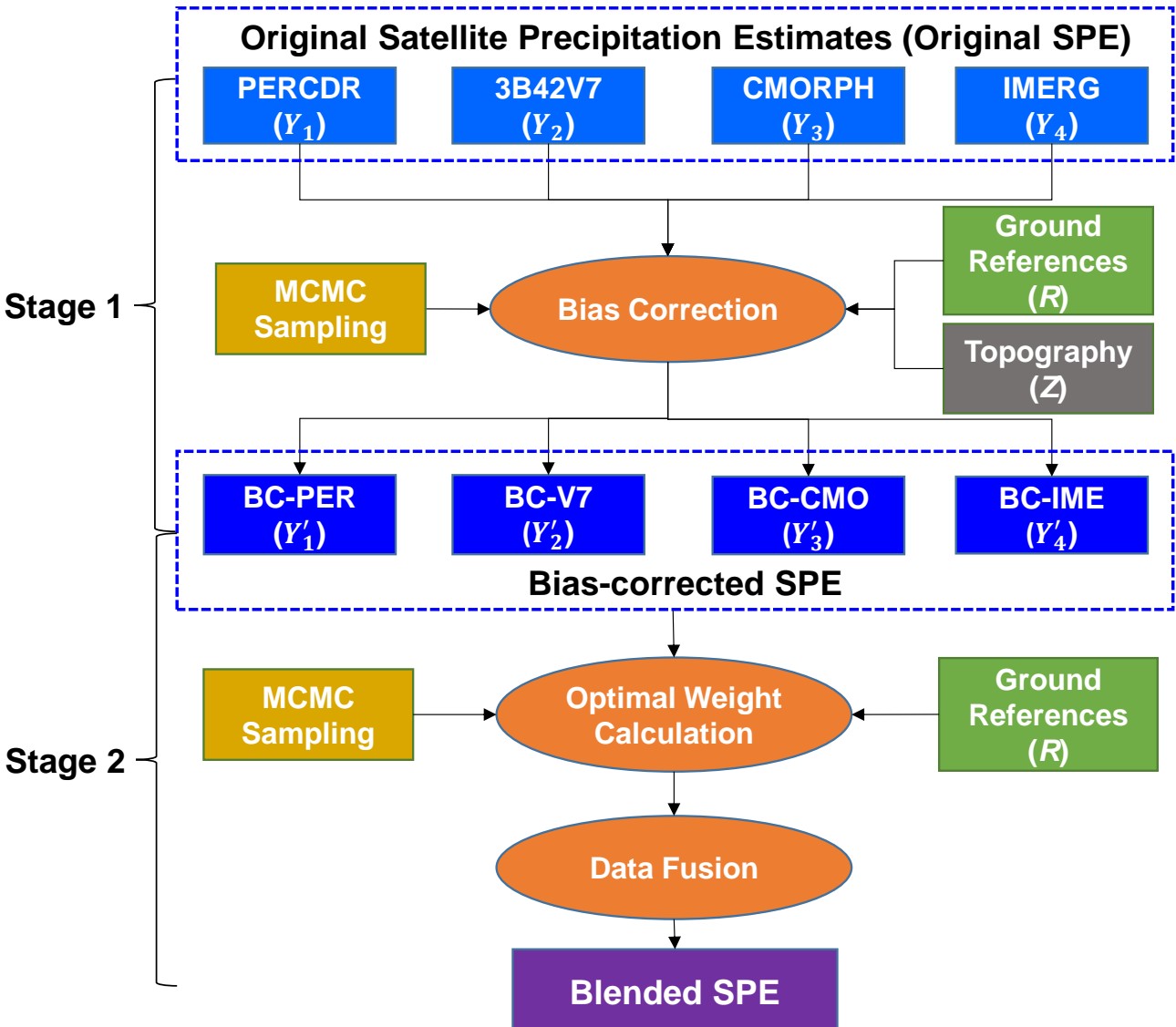

**Figure 2:** The diagram of the proposed TSB algorithm.

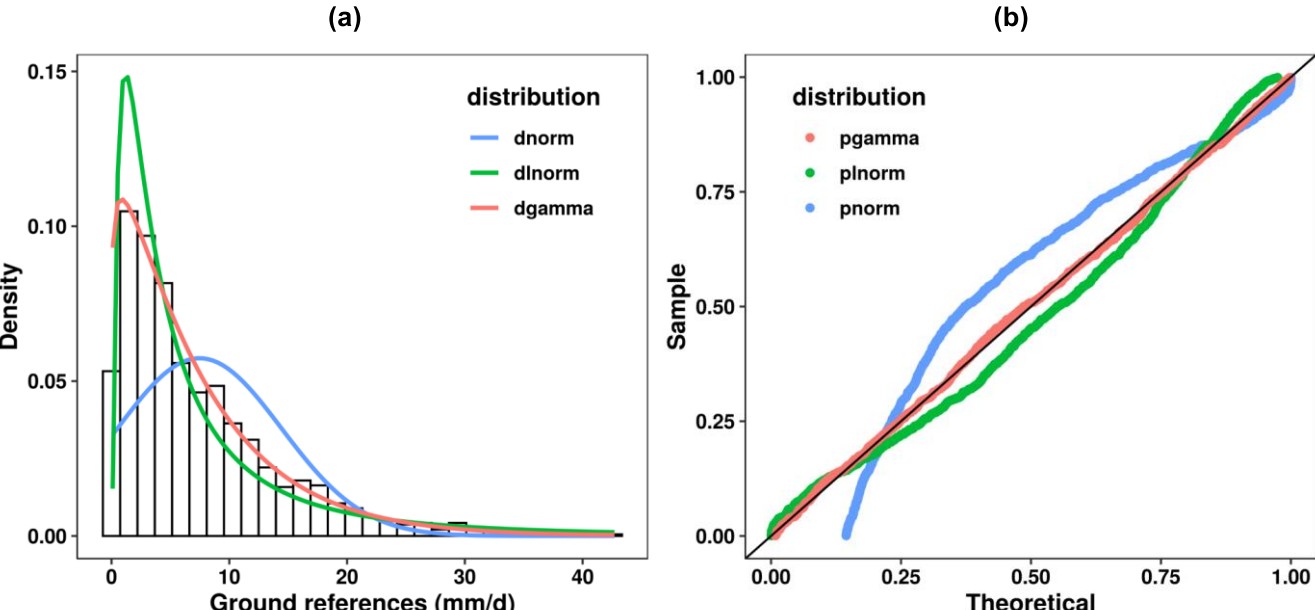

**Figure 3:** (a) The histogram density plot and (b) the corresponding Probability-Probability plot of GR at the training grids in the warm season of 2014 in the NETP, where the red, blue and green lines shows the fitted Gamma, Lognormal and Gaussian distribution, respectively.

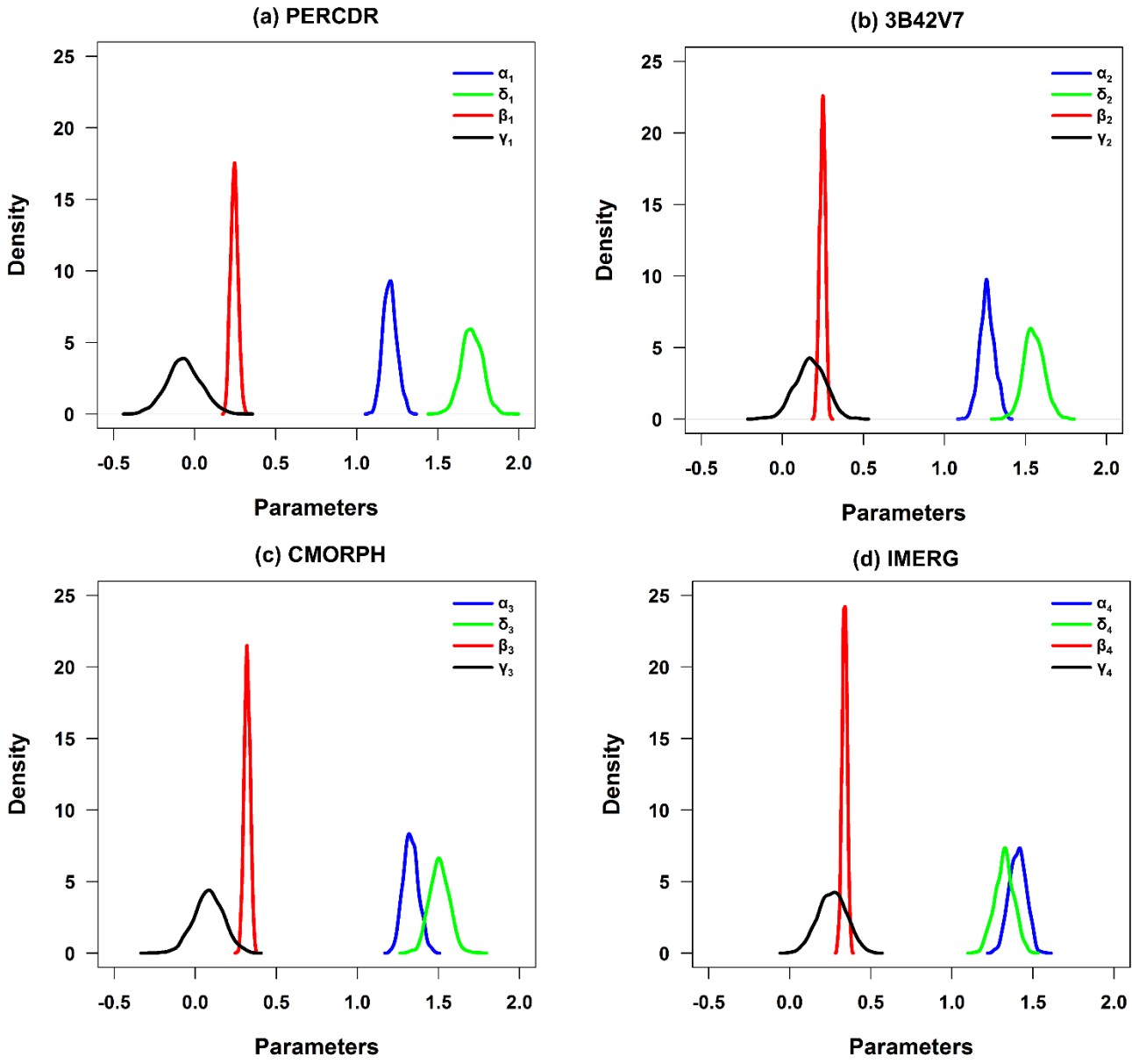

**Figure 4**: The PDF curves of posterior parameter sets with regard to (a) PERCDR, (b) 3B42V7, (c) CMORPH and (d) IMERG

in the bias correction process of Stage 1.

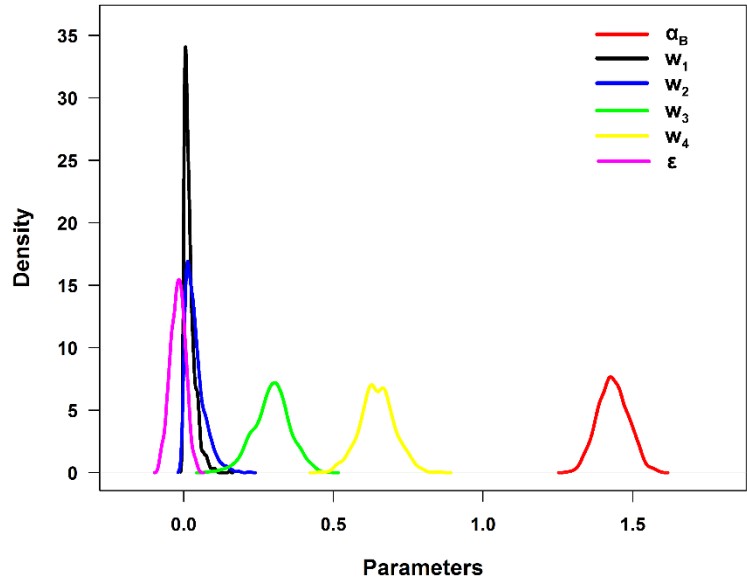

**Figure 5**: The PDF curves of posterior parameter sets in the data fusion process of Stage 2.

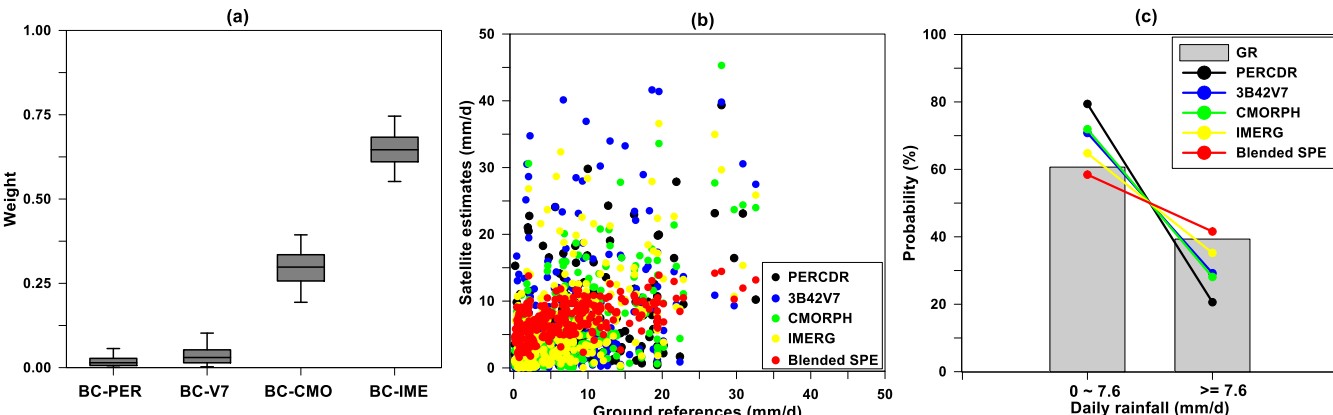

**Figure 6:** (a) The Box-Whisker plots of relative weights for the bias-corrected SPE, (b) the scatter plots between GR and the original SPE and (c) the PDF of daily rainfall for the GR, original and blended SPE with various rain intensities in Scenario 1 in the NETP.

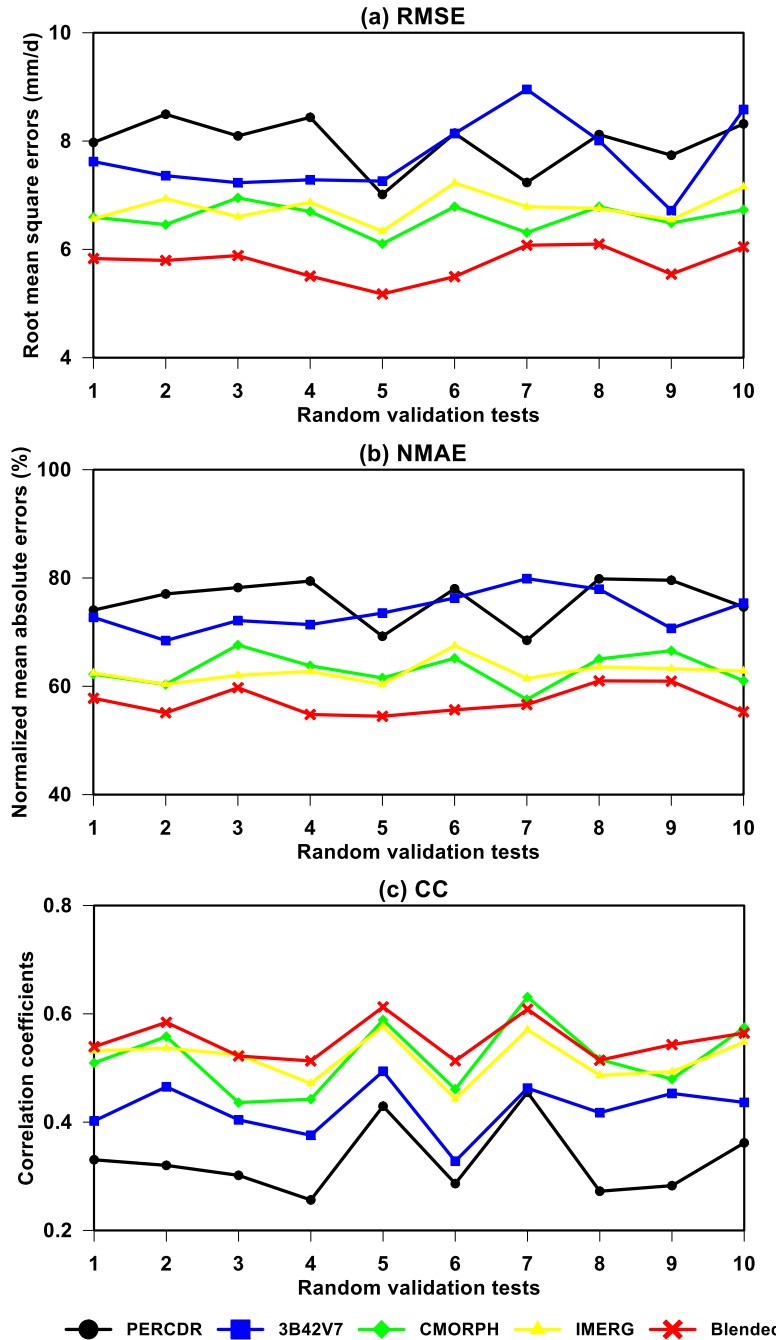

**Figure 7:** Statistical error indices of the original and blended SPE at 10 random verified tests in the warm season of 2014 in the NETP: (a) RMSE, (b) NMAE and (c) CC.

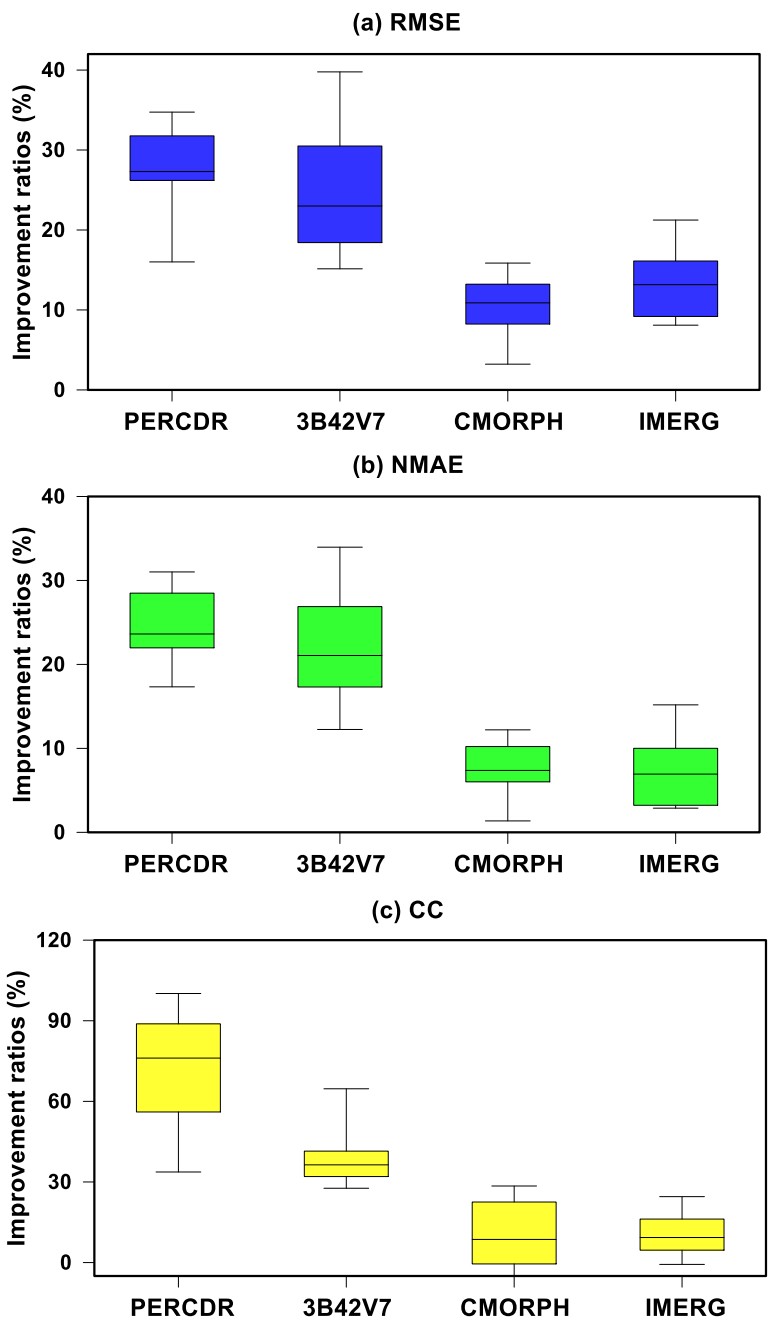

**Figure 8:** The Box-Whisker plots of improvement ratios of statistics for the blended SPE compared with the original SPE, including PERCDR, 3B42V7, CMORPH, and IMERG at 10 random verified tests in the warm season of 2014 in the NETP:
(a) RMSE, (b) NMAE and (c) CC.

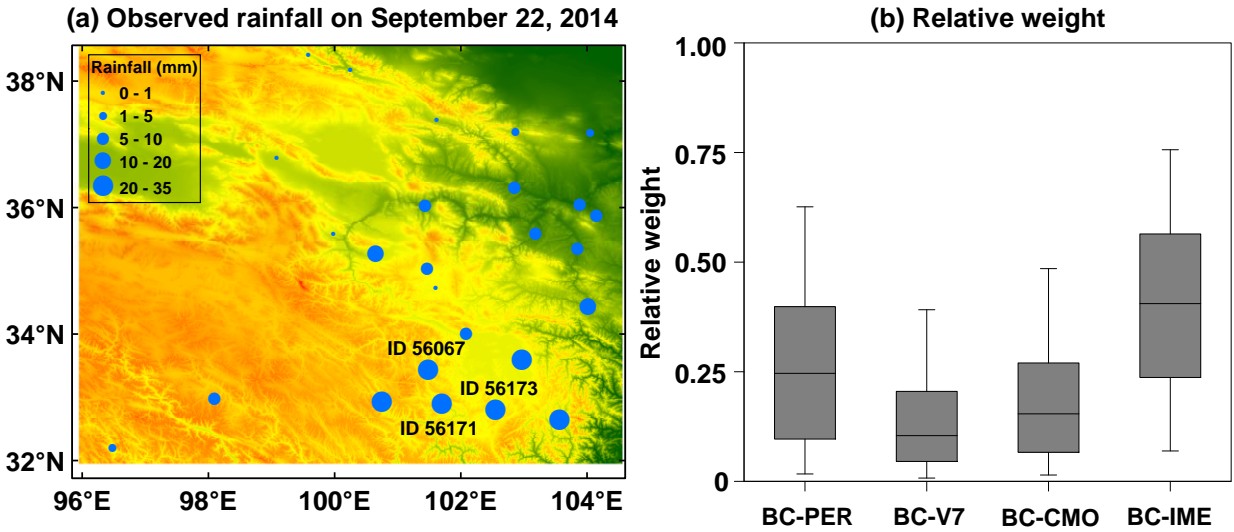

**Figure 9:** (a) Spatial pattern of gauge-based measurements during a heavy rainfall case of September 22, 2014 in the NETP, where the site IDs 56171, 56173 and 56067 report the top three daily rainfall amounts of 32.3 mm, 30.9 mm and 28.7 mm, respectively; (b) the corresponding Box-Whisker plots of relative weights of the bias-corrected SPE in the data fusion process.


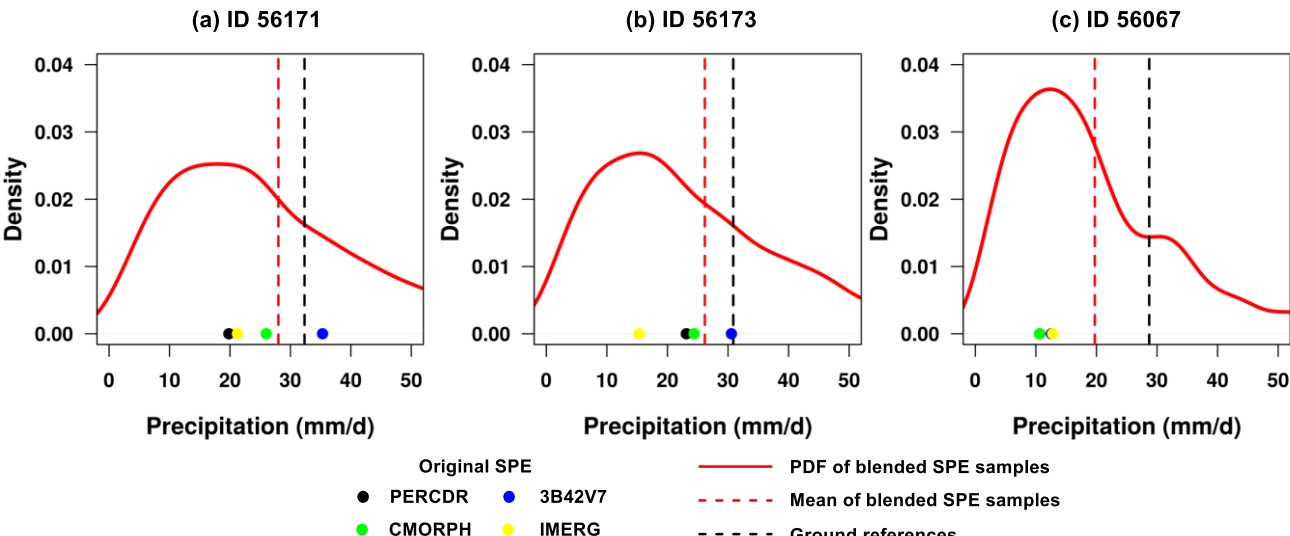

**Figure 10:** The PDF curves of blended SPE samples and the corresponding mean value at three gauge-based grids on a heavy rainfall case of September 22, 2014: (a) ID 56171, (b) ID 56173 and (c) ID 56067. The original SPE and GR at each pixel are also indicated in each subfigure.


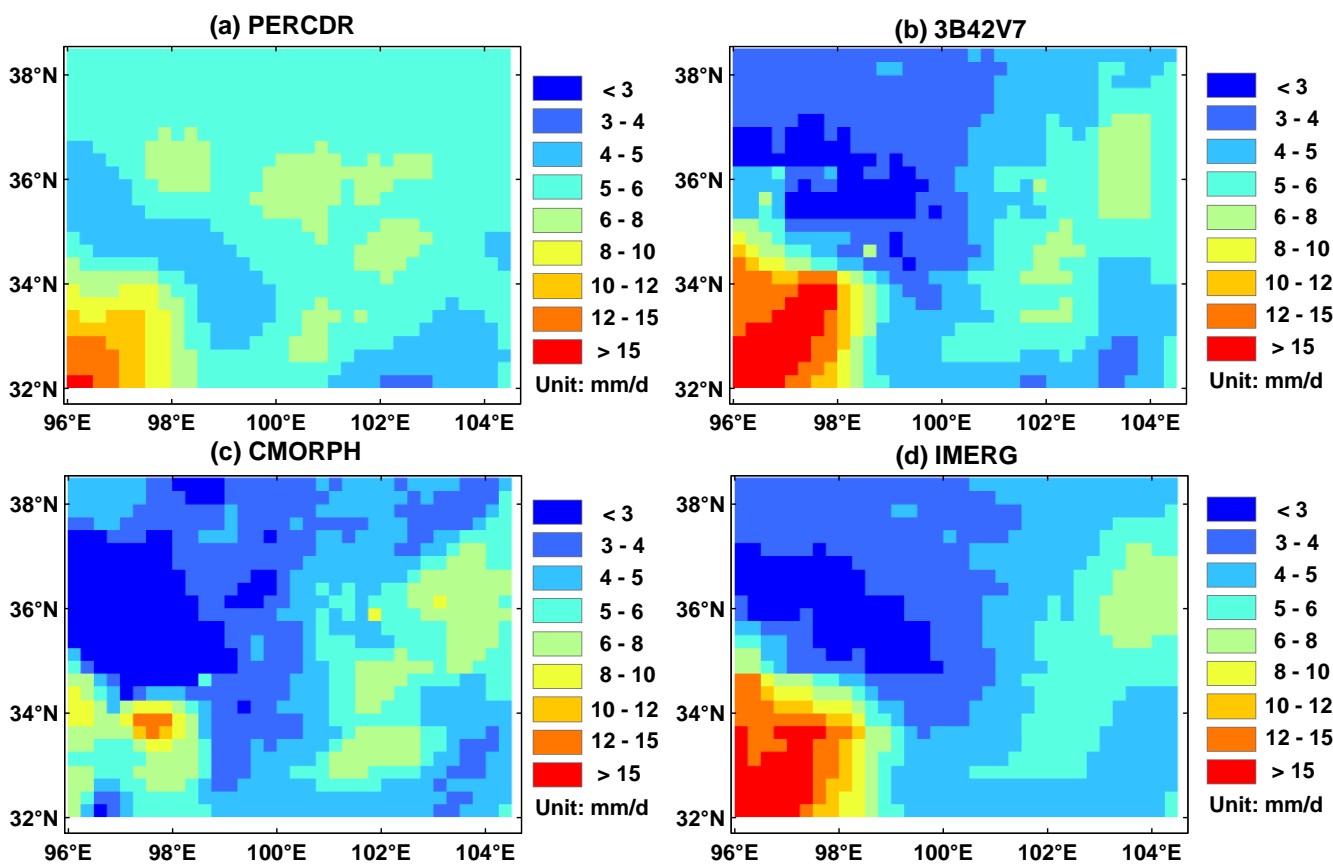

**Figure 11:** Spatial patterns of the daily mean precipitation in terms of the original SPE in the warm season of 2010 to 2014 in the NETP: (a) PERCDR, (b) 3B42V7, (c) CMORPH, and (d) IMERG.

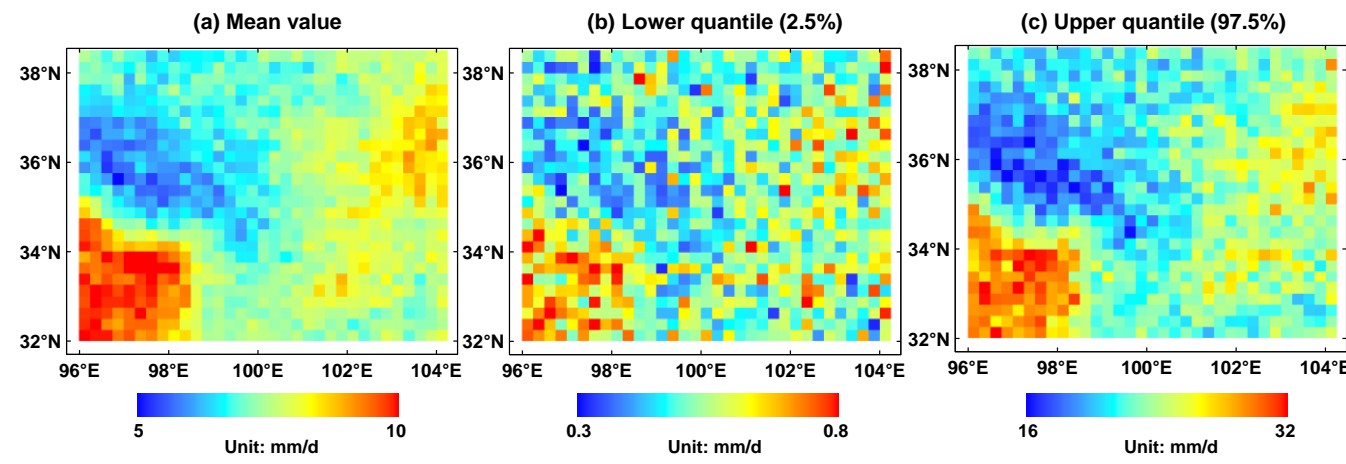


**Figure 12:** Spatial patterns of the blended SPE in terms of (a) mean, (b) lower quantile (2.5%) and (c) upper quantile (97.5%) of daily mean precipitation in the warm season of 2010 to 2014 in the NETP.

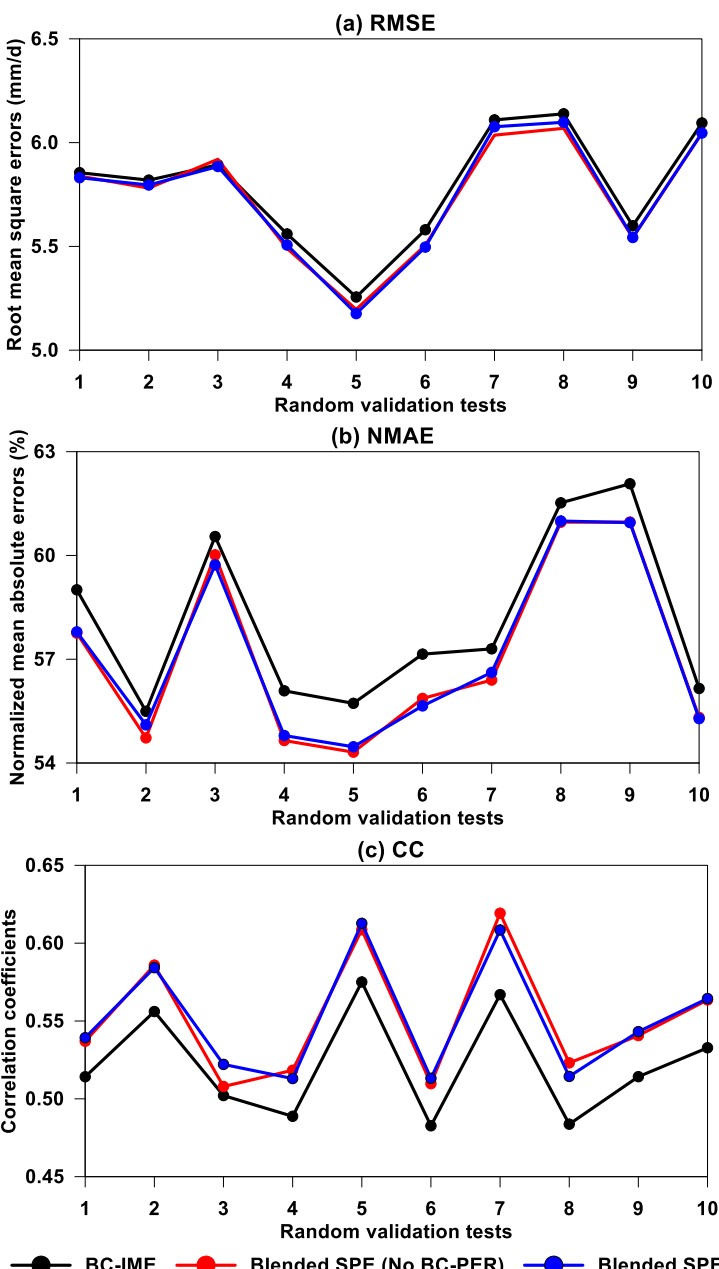

**Figure 13:** Statistical error indices (i.e., RMSE, NMAE, and CC) of the best-performed bias-corrected SPE (i.e., BC-IME, black) and blended SPE before (red) and after (blue) removing the worst-performed BC-PER at 10 random verified tests in the warm season of 2014 in the NETP.

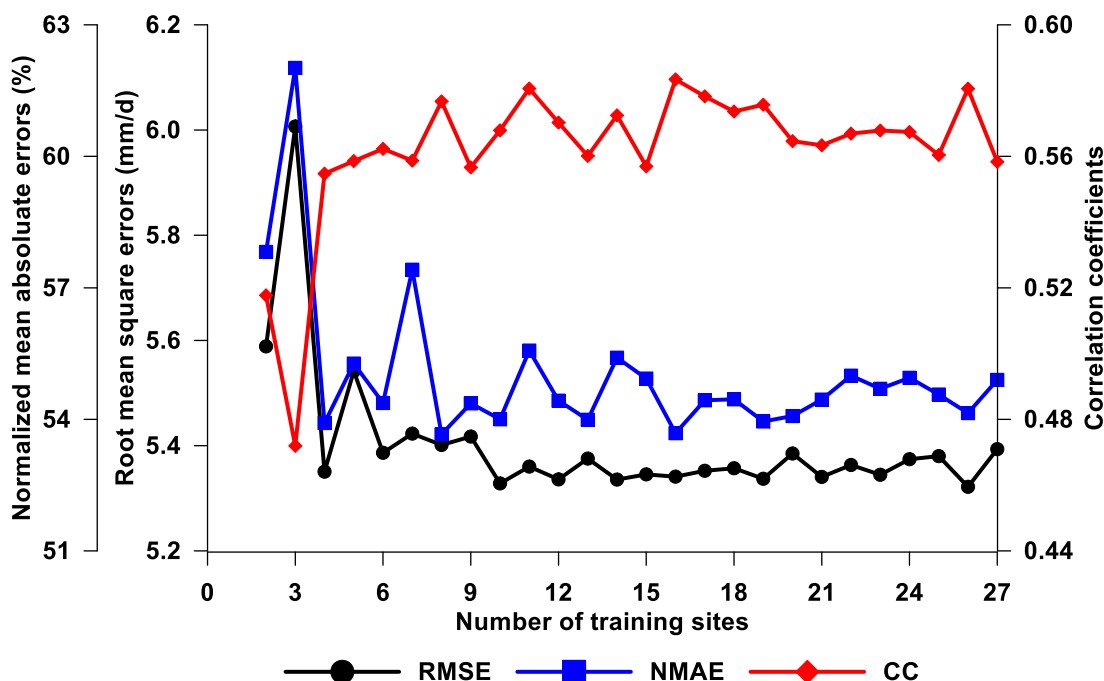

**Figure 14:** Statistical error indices (i.e., RMSE, NMAE, and CC) of the blended SPE at the validation grid locations in terms of different number of training sites in the warm season of 2014 in the NETP.