# Peer review of "A two-stage blending approach for merging multiple satellite precipitation estimates and rain gauge observations: An experiment in the northeastern Tibetan Plateau"

_Hydrology and Earth System Sciences, 2020_

## Referee Comment (RC1) · Anonymous Referee #1 · 12 Mar 2020

It is of importance for the scientific community to improve the retrieval accuracy of satellite precipitation estimates over complex terrains. This study proposed a flexible two-step approach to reduce the systematic errors of currently mainstream satellite precipitation products in the northeastern Tibetan Plateau. Evaluation results show that this approach effectively reduce the errors and biases of satellite retrievals. Overall, the paper is rich in content and technically sound. It can offer insightful references for both satellite precipitation produces and data users, especially for improving the retrieval algorithms over mountainous regions. I consider it is clearly written and informative,

and it should be of interest to a significant subset of HESS readers. Thus, I recommend it be accepted for publication, with just a few minor revisions. First, I wonder why the new approach can effectively reduce the biases but not change the CC values. In the text, the authors should explain this point in more details. Second, the study area is limited within a squared rectangle. In practice, it is difficult to present the application potentials of new approach using such relatively small region as study domain (only like a case study). The gauge numbers are still not enough for validation. At least, the authors should discuss this in the section of conclusion. Last but not at least, this manuscript needs to further polish before publication.

---

## Referee Comment (RC2) · Anonymous Referee #2 · 16 Mar 2020

This manuscript describes a two-step methodology to combine multiple satellite precipitation products to produce a blended daily precipitation estimate. The process involves first bias correcting the individual satellite QPE products relative to surface rain gauges. Then, a Bayesian weighting is applied to blend the various QPE datasets into a single product. The approach is demonstrated on a small area in the northeastern part of the Tibetan Plateau over the 2014 warm season, as well as an individual heavy rain case. Overall the manuscript needs to be checked for correct grammar and usage, and the data and methods sections could be lengthened a bit to make things clearer and

therefore reproducible (some specific suggestions for this below). Generally, with a few tweaks to the writing I feel this is publishable with minor revisions.

Specific Comments:

The manuscript would be much easier to follow if consistent terminology were used to refer to original SPE, bias corrected SPE, and blended SPE throughout.

Lines 75-80: Additional information about the data used is needed: Please specify the versions of IMERG and CMORPH you are using, and whether the IMERG is the near real time early, near real time late, or research/final runs. It's also interesting that you chose to use TMPA, which is no longer being produced and is generally very similar to IMERG. Additionally, IMERG, CMORPH, and TRMM-3B42 all have daily products available – why did you choose to use the 3-h products and (presumably) accumulate to daily? Finally, what method did you use to resample the IMERG?

Line 85: If you are using CMORPH V1.0, it also corrects using GPCP.

Line 116: This equation would be easier to read if separated onto 3 lines.

Line 162-170: Some discussion of the effects of comparing point data to somewhat low resolution gridded data is needed.

Line 182-183: It seems that the scatter is reduced for the blended product, but it has induced a high bias for low rain days and a low bias on heavy rain days. It's difficult to see if the bias is improved compared to the original SPE products.

Line 213: I disagree with this statement. PERCDR is clearly very different from the others, and to this point in the manuscript has shown very little value to be kept in consideration, and I think it is worth acknowledging this, then using the case study to point out that PERCDR can in fact be informative and on a case by case basis.

---

## Referee Comment (RC3) · Anonymous Referee #3 · 19 Mar 2020

The manuscript by Ma et al. presents a very interesting study on blending multiple satellite estimates to obtain a better precipitation estimates, especially over region with complex terrain. The analysis is systematic and results support the improvement in precipitation estimates due to two-stage blending approach. During my read, on several occasion I kept searching for necessary details. Unless those details are provided, it is hard to fully evaluate the merit of this work. Therefore I would suggest major revision of the current version of the manuscript. Authors may want to improve the manuscript along the following lines:

[1] Please provide full details of bias adjustment and data merging stages. With the help of some example dataset, Authors need to describe how Equations [1], [2a] and [2b] adjust the bias. Similarly please demonstrate with some dataset how weight parameters were obtained from Equation [3].

[2] Please include plots justifying why Student's t distribution was selected. I am sure at different training sites, different distributions (Lognormal, Gamma, etc.) may show better performance.

[3] Please explain how the information from gridded data (Satellite estimates) was transferred to point locations (training and validation sites). Did Authors apply some downscaling approach? Bringing information from 25km grid to a point in a complex topographical region is challenging.

[4] In equation [1], normalized elevation is used as a covariate. If it is not included, how it will affect the result. Can you quantify it? Was that included just because you are dealing with TP? In the discussions (Section 5), Authors mention about the importance of including other covariates related to precipitation generation mechanism.

[5] As mentioned in Section 2, the data of only warm period from May to September 2014 has been used in this study. Since all the satellite data are available for several years, can Authors perform similar analysis for few years and validate their approach?

[6] Since similar approaches have been developed previously (as mentioned at the end of second paragraph of page 2, Authors should compare the results with the existing approach. The only unique feature of the current approach is that it provides predictive uncertainty.

[7] The results presented in Figures 3 and 4 homogenizes many things. In Figure 3, are you presenting the average value over all the validation sites? I am sure results will differ significantly if you look into individual sites. Also time series plots would show more features than the bar plot. The results from blended is similar to many adjusted

SPE, then can it be concluded that there is no need to blend. Simply apply the stage 1, bias adjustment, and select the best SPE.

[8] In Figure 4, Authors conclude that the blended data have been dropped towards the gauge references but please look at the precipitation with higher values. It appears that red dots have narrow spread for the lower values but SPE is over estimating the values.

[9] Authors claim that the two-stage approach has advantage of not getting impacted by the poor quality SPE. Based on Figure 4a, it can be argued that why to include those SPE which has very low weight. Please justify. Furthermore, Figure 6 shows improvement ratio, of course the SPE with very low weight will show high value here. Why not be careful at the first place in selecting a set of SPE?

[10] Authors talk about CC for a rainfall event (Sept 22, 2014)? Given that the analysis is performed on daily data, how do you obtain CC?

[11] Title says 'A flexible two-stage approach....' In the second paragraph of Section 6, Authors talk about what is the flexibility here. The statement is very general that it is capable of involving a group of multi-SPE. Is that so unique? Please look into it and accordingly modify the title.

[12] Figure 8a is quite different from Figures 7a to 7d. By blending. higher values disappeared from the map except in Southwest corner. Please explain.

[13] The blending product will be extremely beneficial for the areas where there is no or very few rain gauges (specially in mountainous area). However the study area was carefully selected in such a way that the rain gauge density is high. Can the results be extrapolated from the training and validation sites to get the improved blended gridded product, the way Authors have done in Figure 8? If yes, then there must be some guideline how many minimum training sites do I need to apply this two-stage approach in other complex regions.

[12] The manuscript should be thoroughly checked for grammar and usages.

---

## Author Comment (AC1) · 14 Apr 2020

Anonymous Referee #1: It is of importance for the scientific community to improve the retrieval accuracy of satellite precipitation estimates over complex terrains. This study proposed a flexible two-step approach to reduce the systematic errors of currently mainstream satellite precipitation products in the northeastern Tibetan Plateau. Evaluation results show that this approach effectively reduce the errors and biases of satellite retrievals. Overall, the paper is rich in content and technically sound. It can offer insightful references for both satellite precipitation produces and data users, es-

pecially for improving the retrieval algorithm over mountain regions. I consider it is clearly written and informative, and it should be of interest to a significant subset of HESS readers. Thus, I recommend it be accepted for publication, with just a few minor revisions.

Response: We thank this reviewer for the supportive comment.

First, I wonder why the new approach can effectively reduce the biases but not change the CC values. In the text, the authors should explain this point in more details.

Response: As bias correction is performed for each SPE in the first Stage, the blended SPE has a low bias compared with the original SPE. We agree that the CC index does not improve significantly compared to the RMSE and NMAE values for the blended SPE. The CC between two data sets is a measure of how well they are related. In Stage 1, the mean parameter in the Student's t distribution is expressed as a linear regression of the original SPE. A linear assumption in the proposed model might fail to expect significant difference in the correlation. Thus, the other error indices (i.e., RMSE and NMAE) are adopted together to evaluate the performance of the proposed blending approach. We will give more explanations in the revised manuscript as suggested by this reviewer.

Second, the study area is limited within a squared rectangle. In practice, it is difficult to present the application potentials of new approach using such relatively small region as study domain (only like a case study). The gauge numbers are still not enough for validation. At least, the authors should discuss this in the section of conclusion.

Response: We thank this reviewer for the important comments. Perhaps we might not describe them very clearly in the original manuscript. We will rephrase the concerned issues in the revised manuscript as pointed out by this reviewer.

Last but not at least, this manuscript needs to be further polish before publication.

Response: To be polish as suggested!

---

## Author Comment (AC2) · 14 Apr 2020

Anonymous Referee #2: This manuscript describes a two-step methodology to combine multiple satellite precipitation products to produce a blended daily precipitation estimate. The process involves first bias correcting the individual satellite QPE products relative surface rain gauges. Then, a Bayesian weighting is applied to blend the various QPE datasets into a single product. The approach is demonstrated on a small area in the northeastern part of the Tibetan Plateau over the 2014 warm season, as well as an individual heavy rain case. Overall the manuscript needs to be checked for

correct grammar and usage, and the data and methods sections could be lengthened a bit to make things clearer and therefore reproducible (some specific suggestions for this below). Generally, with a few tweaks to the writing I feel this is publishable with minor revisions.

Response: We thank this reviewer for the great comments. The manuscript will be carefully checked to avoid grammar and usage typos. The data and methods sections will be lengthened in the revised manuscript as required by this reviewer.

Specific Comments: The manuscript would be much easier to follow if consistent terminology were used to refer to original SPE, bias corrected SPE, and blended SPE throughout.

Response: Revise as suggested!

Lines 75-80: Additional information about the data used is needed: Please specify the versions of IMERG and CMORPH you are using, and whether the IMERG is the near real time early, near real time late, or research/final runs. It is also interesting that you chose to use TMPA, which is no longer being produced and is generally very similar to IMERG. Additionally, IMERG, CMORPH, and TRMM-3B42 all have daily products available - why did you choose to use the 3-h products and (presumably) accumulate to daily? Finally, what method did you use to resample the IMERG?

Response: The CMORPH V1.0 research products and the Level 3 IMERG V03 final run products are used in this study. We agree that TMPA is similar to IMERG, but the satellite retrieval algorithm between the two products are different. Considered that TMPA 3B42V7 shows a good performance in the TP, it is selected as an individual in this blending process. It is known that the daily scale of SPE is accumulated from the 3-h (TMPA, CMORPH) or 30-min (IMERG), we admit that we can directly use the daily scale instead of accumulation again from the 3-h products as suggested. The nearest neighbor interpolation is used to resampling the IMERG data. We will also rephrase these issues in the revised manuscript as pointed out by this reviewer.

Line 85: If you are using CMORPH V1.0, it also corrects using GPCP.

Response: Corrected as suggested.

Line 116: This equation would be easier to read if separated into 3 lines.

Response: Separated as suggested.

Line 162-170: Some discussion of the effects of comparing point data to somewhat low resolution gridded data is needed.

Response: We will rephrase this statement as suggested by this reviewer.

Line 182-183: It seems that the scatter is reduced for the blended product, but it has induced a high bias for low rain days and a low bias on heavy rain days. It's difficult to see if the bias is improved compared to the original SPE products.

Response: We thank this reviewer for the comment. Yes, the scatter is reduced for the blended SPE. We perform an additional comparison at the validated locations based on various rainfall intensities in Fig. 1. Based on the two-stage blending (TSB) method, the blended SPE have been effectively dropped towards GR at the validation sites (Figure 1b), especially for the rain intensity values less than 15 mm/d (Figure 1c). Also, there is an overestimation for the original SPE but an underestimation for the blended SPE as the daily rainfall is more than 15 mm, partly because the Bayesian correction (BC) process might over-correct the original SPE on the heavy rainfall in this case. Overall, this TSB method has its ability to exert benefits from SPE in terms of higher performance and mitigate poor impacts from the ones with lower quality.

Line 213: I disagree with this statement. PRECDR is clearly very different from the others, and to this point in the manuscript has shown very litter value to be kept in consideration, and I think it is worth acknowledging this, then using the case study to point out that PRECDR can in fact be informative and on a case by case basis.

Response: We fully agree with this reviewer for the comments. We will add the statements in the revised manuscript as pointed by this reviewer.

[Figure]

**Fig. 1.** (a) The Box-Whisker plots of relative weights of SPE; (b) Scatter plots between GR and various SPE; (c) The PDF of the GR, original and blended SPE with various intensities

---

## Author Comment (AC3) · 15 Apr 2020

Anonymous Referee #3:

The manuscript by Ma et al. Presents a very interesting study on blending multiple satellite estimates to obtain a better precipitation estimates, especially over region with complex terrain. The analysis is systematic and results support the improvement in precipitation estimates due to two-stage blending approach. During my read, on several occasion I kept searching for necessary details. Unless those details are provided,

it is hard to fully evaluate the merit of this work. Therefore I would suggest major revision of the current version of the manuscript.

Response: We thank this reviewer for the critical comments. More details of the two-stage blending (TSB) approach is attached in the supplement. We will rephrase them in the revised manuscript as pointed out by this reviewer.

Authors may want to improve the manuscript along the following lines:

[1] Please provide full details of bias adjustment and data merging stages. With help of some example dataset, Authors need to describe how Equations [1], [2a] and [2b] adjust the bias. Similarly please demonstrate with some dataset how weight parameters were obtained from Equation [3].

Response: The details of bias adjustment and data merging stages are provided in the supplemented file. As for the calculation details of Eqs. (1) to (3), please see the supplementary materials. We will also rephrase the Method section in the revised manuscript as pointed out by this reviewer.

[2] Please include plots justifying why Student's t distribution was selected. I am sure at different training sites, different distributions (Lognormal, Gamma, etc.) may show better performance.

Response: We fully agree with this reviewer that at different training sites, different distributions (Lognormal, Gamma, etc.) may show better performance. Given various SPE at different training sites, the specific probabilistic function is not limited to a certain distribution. For demonstration purposes, we herein apply the Student's t distribution, with its mean parameter expressed as a linear regression of the original SPE and terrain feature in this case. We will double check the assumption of Student's t distribution as pointed out by this reviewer.

[3] Please explain how the information from gridded data (Satellite estimates) was transferred to point locations (training and validation) sites. Did Authors apply some

downscaling approach? Bringing information from 25km grid to a point in a complex topographical region is challenging.

Response: Perhaps we didn't describe it very clearly in the original manuscript. We admit that there is a scale gap between gridded SPE and point-based gauge observations. The downscaling approach is not applied in this study. To ensure the same resolution among the original SPE, the IMERG data are resampled from $0.10°$ to $0.25°$ using the nearest neighbor interpolation to eliminate the scale difference. The rain gauge network is spatially interpolated with a $0.25° \times 0.25°$ resolution in the region of interest on each rainy day using a bilinear interpolation approach. The 34 grid cells with the gauge sites are assumed as ground references (GR) in the blending process. We fully agree that it is challenging to bring information from 25 km grid to a point in a complex terrain region. We will rephrase the statements and add some discussions in the revised manuscript as pointed out by this reviewer.

[4] In equation [1], normalized elevation is used as a covariate. If it is not included, how it will affect the result. Can you quantify it? Was that included just because you are dealing with TP? In the discussions (Section 5), Authors mention about the importance of including other covariates related to precipitation generation mechanism.

Response: We are thankful to this reviewer for the important comment. We quantify the impact of elevation covariate on the bias-corrected and blended SPE performances as pointed out by this reviewer in Table 1. It is found that the consideration of elevation performs slightly better skill compared with the model without terrain information in this case study. We would like to admit that it is an initial exploration partly because we are dealing with the TP. We will rephrase these concerns in the revised manuscript as pointed out by this reviewer.

[5] As mentioned in Section 2, the data of only warm period from May to September 2014 has been used in this study. Since all the satellite data are available for several years, can Authors perform similar analysis for few years and validate their approach?

Response: We thank the reviewer for this suggestion. Please allow us to explain that this study aims to develop a newly TSB algorithm on the multi-satellite precipitation data fusion in a certain time in regions of interest. Given that the larger challenge in the TP is to provide more accurate rainfall in a spatial domain, we are trying to overcome the shortage of limited rain gauge network based on the available SPE with spatial advantage using the TSB method in the NETP as a demonstration purpose. We agree that the satellite data are available for several years, but the exploration of long-term periods for the TSB method is another critical issue, e.g., the consideration of time impact on the fusion result. We will replenish this discussion in the revised manuscript. Additionally, the model performance of this new approach is demonstrated based on various aspects, and the evaluation analysis for long-term period and extended regions (e.g., TP) will be addressed in the future work.

[6] Since similar approaches have been developed previously (as mentioned at the end of second paragraph of page 2, Authors should compare the results with the existing approach. The only unique feature of the current approach is that it provides predictive uncertainty.

Response: We thank the reviewer for this great suggestion. It is very important to formally quantify the predictive uncertainty in the Bayesian analysis, which is one of the unique features for the TSB method. As suggested by this reviewer, the TSB approach is compared with two existing fusion approach, i.e., Bayesian model averaging (BMA) and One-outlier removed (OOR) in Table 2. The statistical summary of data comparison among the three fusion approaches at the validated locations are shown below. The TSB approach performs the best skill with the RMSE, NMAE and CC at 4.34 mm/d, 49.2%, and 0.606, respectively, as compared with the other two fusion methods. We will add this comparison in the revised manuscript as recommended by this reviewer.

[7] The results presented in Figures 3 and 4 homogenizes many things. In Figure 3, are you presenting the average value over all the validation sites? I am sure results will differ significantly if you look into individual sites. Also time series plots would show

more features than the bar plot. The results from blended is similar to many adjusted SPE, then can it be concluded that there is no need to blend. Simply apply the stage 1, bias adjustment, and select the best SPE.

Response: We are thankful to the reviewer for these comments. Yes, Figure.3 in the original manuscript presents the statistical error summary over all the validation sites. We agree that there are more features if looking into individual sites than overall evaluation of the validated sites. The time series plots of daily rainfall estimates and rainfall accumulations in terms of GR, original and blended SPE at a validated grid cell with a rain gauge labeled as ID 56173 is shown in Figure 1 below as a demonstration example. This rain gauge locates in 32.8° N, 102.55°E, 3484 m, and has the maximum rainfall records in the warm season of 2014 in the NETP. Visual analysis shows that the blended SPE provides reasonable rainfall as compared to the original SPE, and the blended SPE also has a better skill in terms of RMSE at 4.95 mm/d compared with the original SPE including PERCDR (10.71), 3B42V7 (9.76), CMORPH (8.0), and IMERG (10.49), respectively.

This reviewer also raises a question that why not be careful at the first place in selecting a good set of SPE, or simply apply the first stage of bias correction and then select the best SPE as the final product. To address this issue, we investigate the error differences among the best-performed SPE, i.e., BC-IME, and blended SPE before and after the removing of the worst-performed SPE, i.e., PERCDR, for 10 random verified tests in the warm season of 2014 in the NETP (Figure 2). It implies that it is beneficial to involve the second stage in the TSB method as the blended SPE performs better skills than the simply bias correction step with the best-performed SPE. The primary reason is that the blended step is designed to integrate various types of SPE, which is limited for the simple bias-corrected step. Also, both blended SPE products in Figure 2 show similar performance in terms of the RMSE, NMAE, and CC indices. The TSB approach has an advantage of not getting impacted by the poor quality SPE in the application, partly because the proposed BW model in Stage 2 can reallocate the contribution of

the SPE based on their corresponding bias characteristics.

We will rephrase the related concerns in the revised manuscript.

[8] In Figure 4, Authors conclude that the blended data have been dropped towards the gauge references but please look at the precipitation with higher values. It appears that red dots have narrow spread for the lower values but SPE is over estimating the values.

Response: We are thankful to the reviewer for the important comment. We also notice that there is an overestimation for the original SPE compared to GR, and the blended SPE shows different spread at various rainfall intensities. To address this issue, we perform an additional analysis of probability density function of daily rainfall with various intensities at the validated locations in Figure 3c. There is an overestimation for the original SPE but an underestimation for the blended SPE as the daily rainfall is more than 15 mm, partly because the BC process might over-correct the original SPE on the heavy rainfall in this case. We will rephrase this statement in the revised manuscript as pointed out by this reviewer.

[9] Authors claim that the two-stage approach has advantage of not getting impacted by the poor quality SPE. Based on Figure 4a, it can be argued that why to include those SPE which has very low weight. Please justify. Furthermore, Figure 6 shows improvement ratio, of course the SPE with very low weight will show high value here. Why not be careful at the first place in selecting a set of SPE?

Response: We thank this reviewer for the critical comment. It is well known that the SPE are obtained from different satellite retrieval algorithms, and each of them can provide various rainfall information. The over-performed SPE would provide more information, and the poor-performed ones give less value. It is thus necessary to integrate all kinds of SPE so as to reduce the predictive uncertainty in the domain. The proposed TSB approach has an advantage of integrating various SPE information and not getting impacted by the poor quality of SPE, partly because the Bayesian weight model

in Stage 2 is can reallocate the contribution of the SPE based on their corresponding bias characteristics.

To address this issue, we also investigate the statistical error difference among the best-performed SPE, and blended SPE before and after the removing of the worst-performed SPE, i.e., PERCDR in this case, for 10 random verified tests in the warm season of 2014 in the NETP (Figure 2). It is found that the blended SPE performs better skill than the simply bias correction with the best-performed SPE (e.g., BC-IME in this case), and both blended SPE products show similar performance with the RMSE, NMAE, and CC indices. It proves that it is beneficial to involve the second stage in the TSB method.

We will rephrase the related expressions in the revised manuscript.

[10] Authors talk about CC for a rainfall event (Sept 22, 2014)? Given that the analysis is performed on daily data, how do you obtain CC?

Response: There are 27 rain gauge sites in total that has a rainfall record on Sep 22, 2014 in the regions of interest. The CC index is calculated based on the data sets from the 27 grid cells.

[11] Title says 'A flexible two-stage approach...' In the second paragraph of Section 6, Authors talk about what is the flexibility here. The statement is very general that it is capable of involving a group of multi-SPE. Is that so unique? Please look into it and accordingly modify the title.

Response: We thank this reviewer for the important comment. The word of "flexible" is removed in the title. We will replace the title with "A two-stage approach for blending multiple satellite precipitation estimates and rain gauge networks: An experiment in the northeastern Tibetan Plateau" in the revised manuscript.

[12] Figure 8a is quite different from Figures 7a to 7d. By blending, higher values disappeared from the map except in Southwest corner. Please explain.

Response: Thank you for this specific comment. Fig.8a is the spatial map of the blended result which is weight summation of the original SPE from Figs. 7a to 7d. There is an overestimation for most of the original SPE in the NETP in this experiment, the bias of the blended SPE is reduced based on the TSB approach. Thus, higher values disappear from the map except in southwest corner. Because daily mean rainfall is the highest in southwest corner for each SPE, higher value exists after the blending process. We will explain this issue in the revised manuscript as pointed out by this reviewer.

[13] The blending product will be extremely beneficial for the areas where there is no or very few rain gauges (specially in mountain area). However the study area was carefully selected in such a way that the rain gauge intensity is high. Can the results be extrapolated from the training and validations sites to get the improved blended gridded product, the way Authors have done in Figure 8? If yes, then there must be some guideline how many minimum training sites do I need to apply this two-stage approach in other complex regions.

Response: We are thankful to this reviewer for the comment. As pointed out by this reviewer, it is helpful to give some guideline that how many minimum training sites are needed to apply the TSB approach in a region with complex terrain and limited ground observations. The sensitivity analysis of the number of training grid cells on the performance of blended SPE at the validated sites is explored in Figure 5. As the number of training sites is increasing, there is a decreasing trend for the RMSE and NMAE values, but a slight increasing trend for the CC value. Except for an anomaly with No. 23, the performance of the blended SPE becomes similar as the number of training sites increases to 21 in this case. Also, it is noted that if more useful information is provided from the involved SPE and rain gauges, it is more beneficial for the blended gridded product in the region of interest. We will rephrase this critical issue in the discussion part in the revised manuscript.

[14] The manuscript should be thoroughly checked for grammar and usages.

Response: To be thoroughly checked as suggested.

Please also note the supplement to this comment:
https://www.hydrol-earth-syst-sci-discuss.net/hess-2020-43/hess-2020-43-AC3-
supplement.pdf

―――――――――――――――――
43, 2020.

[Figure]

[Figure]

**Fig. 1.** Time series of (a) daily rainfall estimates and (b) rainfall accumulations at a selected validation grid location with the maximum rainfall records in the warm season of 2014.

[Figure]

**Fig. 2.** Statistical error indices of the best-performed bias-corrected SPE (i.e., BC-IME, black) and blended SPE before (red) and after (blue) the removing of the worst-performed PERCDR for 10 random tests

[Figure]

**Fig. 3.** (a) The Box-Whisker plots of relative weights of SPE in Stage 2; (b) Scatter plots between GR and various SPE; (c) The PDF of the GR, original and blended SPE with various rainfall intensities

**Fig. 4.** Statistical error indices of the blended SPE at the validated grid locations in terms of different number of training sites in the warm season of 2014 in the NETP

[Figure]

**Table 1:** Summary of statistical error indices (i.e., RMSE, NMAE, and CC) in terms of bias-corrected and blended SPE with and without consideration of terrain feature as a covariate in the TSB method at the validated grid cells of NETP in the warm season of 2014.

| Product | Type | RMSE (mm/d) | NMAE (%) | CC |
|---|---|---|---|---|
| BC-PER | No Terrain | 5.03 | 58.9 | 0.416 |
| | Terrain | 5.02 | 58.7 | 0.418 |
| BC-V7 | No Terrain | 5.08 | 58.0 | 0.403 |
| | Terrain | 5.06 | 57.5 | 0.410 |
| BC-CMO | No Terrain | 4.83 | 55.0 | 0.493 |
| | Terrain | 4.81 | 54.6 | 0.497 |
| BC-IME | No Terrain | 4.58 | 51.4 | 0.568 |
| | Terrain | 4.56 | 50.9 | 0.572 |
| Blended SPE | No Terrain | 4.36 | 49.7 | 0.603 |
| | Terrain | 4.34 | 49.2 | 0.606 |

**Table 2**: Summary of statistical error indices (i.e., RMSE, NMAE, and CC) in terms of

three fusion methods (i.e., OOR, BMA, and TSB) at the validated grid cells of NETP in

the warm season of 2014.

| Method | RMSE (mm/d) | NMAE (%) | CC |
|--------|-------------|----------|-----|
| OOR | 5.63 | 59.2 | 0.547 |
| BMA | 5.44 | 57.6 | 0.595 |
| TSB | 4.34 | 49.2 | 0.606 |

[Figure]

**Supplement:**

5    Yingzhao Ma et al.: yzma@colostate.edu

**Topic:** More details of the two-stage blending (TSB) approach

**3 The TSB algorithm**

**3.1 Overview**

This algorithm aims at developing a multi-source data merging framework to provide the best-available gridded
10   precipitation product with GR and SPE in the region of interest. Let $R(s,t)$ denote near-surface precipitation at the
GR cell $s$ and the $t^{th}$ day. The original SPE and bias-corrected SPE are defined as $(Y_1(s,t), Y_2(s,t), ..., Y_p(s,t)$ and
$(Y_1'(s,t), Y_2'(s,t), ..., Y_p'(s,t))$ at the same grid and time, respectively. For simplicity, they are separately replaced by
$R$, $(Y_1, Y_2, ..., Y_p)$, and $(Y_1', Y_2', ..., Y_p')$. The subscript $p$ implies the number of SPE in terms of its value at 4 in the
following application, and PERCDR, 3B42V7, CMORPH and IMERG refer to $Y_1, Y_2, Y_3, Y_4$, respectively.

15

The diagram of the TSB method is shown in Figure 2. Stage 1 is designed to mitigate the bias of SPE based on the
GR at the training sites with a Bayesian correction (BC) procedure, where the assumption of probabilistic distribution
for GR conditional on each SPE is not limited to Gaussian prototype. Given complex terrain and $0.25°$ grid resolution,
the topography is added as a covariate in the BC process. In the $2^{nd}$ stage, a Bayesian weight (BW) model is used to
20   merge the bias-corrected SPE. The BW model can exert benefit from bias-adjusted SPE with high performance and
reduce poor impact from the ones with lower quality. It also produces blended SPE with predictive uncertainties. The
details of the TSB algorithm are described in Sections 2.2 and 2.3, respectively.

**3.2 Stage 1: Bias correction**

25   In this stage, we perform on conditional modelling of GR on each SPE, i.e., on the probabilistic distribution $f(R)$ at
the training sets to improve the accuracy of the original SPE. A flexible assumption (e.g., Lognormal, Gaussian, or
Student's $t$ distribution) for bias characteristics between GR and SPE is proposed. Given various SPE at different
training sites, the specific probabilistic function is not limited to a certain distribution. For demonstration purposes,

we herein apply the Student's $t$ distribution, with its mean parameter expressed as a linear regression of the original SPE and terrain feature in the case. It is parameterized below:

$$R \sim Student(\nu_i, \mu_i, \sigma_i) \tag{1}$$

$$\mu_i = \alpha_i + \beta_i * Y_i + \gamma_i * Z \tag{2}$$

where $\nu_i$ is known as degree of freedom, $\mu_i$ and $\sigma_i$ stand for sample mean and variance, respectively; the parameter $\mu_i$ is correlated with the intensity value of the $i^{th}$ SPE ($Y_i$) and terrain feature (Z). To ignore data anomaly, the elevation feature in Eq. (2) is normalized with its value ranging from 0 to 1 in the model application. $\boldsymbol{\theta} = \{\nu_i, \alpha_i, \beta_i, \gamma_i, \sigma_i\}$ is summarized as parameter sets. It further enables to write the likelihood function or probability density function (PDF) from Eqs. (1) and (2) conditional on $\boldsymbol{\theta}$ and $Y_i$ as:

$$f(R|\boldsymbol{\theta}, Y_i) = \frac{\Gamma((\nu_i+1)/2)}{\Gamma(\nu_i/2)} \frac{1}{\sqrt{\nu_i \pi}\,\sigma_i} (1 + \frac{1}{\nu_i}(\frac{R-(\alpha_i+\beta_i*Y_i+\gamma_i*Z)}{\sigma_i})^2)^{-(\nu_i+1)/2} \tag{3}$$

According to the Bayes's theorem (Gelman et al., 2013), the posterior distribution of parameter sets $\boldsymbol{\theta}$ given GR and SPE data, and the prior distribution of parameters $f(\boldsymbol{\theta})$ can be expressed as:

$$f(\boldsymbol{\theta}|R, Y_i) \propto f(R|\boldsymbol{\theta}, Y_i)f(\boldsymbol{\theta}) \tag{4}$$

The estimation of the posterior distribution $f(\boldsymbol{\theta}|R, Y_i)$ in Eq. (4) is challenging as its dimension grows with the number of parameters (Renard, 2011). Here, the Markov Chain Monte Carlo (MCMC) technique complied in the Stan programming language is used to address this issue (Gelman et al., 2013). Given that the assumption of the weakly informative priors ensures the Bayesian inferences in an appropriate range (Ma et al., 2020), the priors of $f(\boldsymbol{\theta})$ are initialized as uniform distribution with $\alpha_i, \beta_i, \gamma_i$ at real numbers in Eq. (5), and with $\nu_i, \sigma_i$ at a lower-bound zero of real numbers in Eq. (6).

$$\alpha_i, \beta_i, \gamma_i \sim Uniform(-\infty, +\infty) \tag{5}$$

$$\nu_i, \sigma_i \sim Uniform(0, +\infty) \tag{6}$$

Based on the estimated parameter sets $\boldsymbol{\theta}$ above, the next step is to calculate the bias-corrected SPE $R^*$ at any new site. It can be quantitatively simulated from its posterior distribution in Eq. (7) using the original SPE $Y_i^*$, and training data $R, Y_i$:

$$f(R^*|Y_i^*, R, Y_i) = \int f(R^*, \boldsymbol{\theta}|Y_i^*, R, Y_i)\, d\boldsymbol{\theta} \tag{7}$$

Following the rule of joint probabilistic distributions, the right term inside the integral of Eq. (7) is written as:

$$f(R^*, \boldsymbol{\theta}|Y_i^*, R, Y_i) = f(R^*|Y_i^*, R, Y_i, \boldsymbol{\theta})f(\boldsymbol{\theta}|Y_i^*, R, Y_i) \tag{8}$$

Given that $Y_i^*$ is independent with $R$ and $Y_i$, the first term of the right side in Eq. (8) is transformed as:

$$f(R^*|Y_i^*, R, Y_i, \boldsymbol{\theta}) = f(R^*|Y_i^*, \boldsymbol{\theta}) \tag{9}$$

Since the parameters $\boldsymbol{\theta}$ are dependent upon the training data $R, Y_i$, the second term of the right side in Eq. (8) is expressed as:

60

$$f(\boldsymbol{\theta}|Y_i^*, R, Y_i) = f(\boldsymbol{\theta}|R, Y_i) \tag{10}$$

Therefore, the posterior predictive distribution of $R^*$ in Eq. (7) is written below:

$$f(R^*|Y_i^*, R, Y_i) = \int f(R^*|Y_i^*, \boldsymbol{\theta}) f(\boldsymbol{\theta}|R, Y_i) \, d\boldsymbol{\theta} \tag{11}$$

Since there is no general way to calculate the associated integral in Eq. (11), it is performed again using the MCMC iterations. A numerical algorithm is suggested below: $n_{sim}$ is assumed as the replicates of the post-convergence

65   MCMC samples, and the predicted samples for $R^*$ in Eq. (11) is iterated ($i = 1, \ldots, n_{sim}$) as follows:

1) Calculate the model parameters $\boldsymbol{\theta}$ from Eqs. (1) to (6) described above;

2) Compute the mean parameter $\mu_i^*$ from the regression model of Eq. (2), i.e., $\mu_i^* = \alpha_i + \beta_i * Y_i^* + \gamma_i * Z^*$;

3) Generate the derived quantity from the posterior distribution of $R^*$ in Eq. (11).

70   **3.3 Stage 2: Data merging**

On the basis of Stage 1 in Section 3.2, the median value of the posterior samples is used as the bias-corrected SPE. Here, we redefine the bias-corrected SPE as $Y_i'$ ($i = 1,2,\ldots,p$). The formulas of blending the bias-adjusted SPE are shown below:

$$B = \sum_{i=1}^{p} Y_i' * w_i + \varepsilon \tag{12}$$

75

$$\sum_{i=1}^{p} w_i = 1 \tag{13}$$

$$\varepsilon \sim Normal(0, \sigma) \tag{14}$$

$$w_i \sim Uniform(0,1), i = 1, \ldots, p \tag{15}$$

$$\sigma \sim Uniform(0, +\infty) \tag{16}$$

where $B$ means the blended SPE; $w_i$ ($i=1,2,\ldots,p$) stands for the relative weight of the $i^{th}$ bias-corrected SPE with its

80   value ranging from 0 to 1; $\varepsilon$ is the residual error with its value at positive real number. Ideally, the blended SPE at the training site $s$ and time $t$ are close to GR, i.e., $R(s, t)$. Thereby, model parameters $\boldsymbol{\delta}$, including $w_i (i = 1,2,\ldots p)$ and $\sigma$

will be estimated based on GR and bias-corrected SPE at the training sites. With regard to the conditional distribution of blended SPE on the bias-corrected SPE, we propose a Gaussian distribution for residual error modelling. The corresponding PDF is written as follows:

85

$$f(B|\boldsymbol{\delta}) = \frac{1}{\sqrt{2\pi}\sigma} \exp(-\frac{1}{2}(\frac{B-\sum_{i=1}^{p} Y_i' * w_i}{\sigma})^2) \tag{17}$$

The calculation process of $\boldsymbol{\delta}$ is similar with the parameter estimation described in Stage 1. After the parameters $\boldsymbol{\delta}$ are estimated, similar to Eqs. (7) to (11), the blended SPE at any site and time $t$ can be derived with the bias-corrected SPE and corresponding weights using the MCMC iterations. Finally, we can obtain spatial patterns of blended SPE in terms of the median, standard deviation (SD) and associated credible intervals (e.g., 5% and 95% quantiles) in regions

90    of interest.

---

## Author Response (AR1)

**Responses to Review Comments on HESS-2020-43**

We are thankful to the three anonymous reviewers for their comments and suggestions. We have addressed the comments point by point in the revision. In the text below, comments are repeated verbatim and the corresponding responses are in blue. In addition, we have made substantial improvements to the manuscript based on the comments and suggestions.
* * *
**Anonymous Referee #1:**

It is of importance for the scientific community to improve the retrieval accuracy of satellite precipitation estimates over complex terrains. This study proposed a flexible two-step approach to reduce the systematic errors of currently mainstream satellite precipitation products in the northeastern Tibetan Plateau. Evaluation results show that this approach effectively reduce the errors and biases of satellite retrievals. Overall, the paper is rich in content and technically sound. It can offer insightful references for both satellite precipitation produces and data users, especially for improving the retrieval algorithm over mountain regions. I consider it is clearly written and informative, and it should be of interest to a significant subset of HESS readers. Thus, I recommend it be accepted for publication, with just a few minor revisions.

**Response:** We thank this reviewer for the supportive comment.

First, I wonder why the new approach can effectively reduce the biases but not change the CC values. In the text, the authors should explain this point in more details.

**Response:** As bias correction is performed for each SPE in the first stage, the blended SPE has a low bias compared with the original SPE. We agree that the CC index does not improve significantly compared to the RMSE and NMAE values for the blended SPE. The CC between two data sets is a measure of how well they are related. In Stage 1, the mean parameter in the Student's $t$ distribution is expressed as a linear regression of the original SPE. A linear assumption in the proposed model might fail to expect significant difference in the correlation. Thus, the RMSE and NMAE indices are also adopted to evaluate the performance of the proposed blending approach. We have given more explanations in the revised manuscript as suggested by this reviewer (Lines 190-192).

Second, the study area is limited within a squared rectangle. In practice, it is difficult to present the application potentials of new approach using such relatively small region as study domain (only like a case study). The gauge numbers are still not enough for validation. At least, the authors should discuss this in the section of conclusion.

**Response:** We thank this reviewer for the important comments. Perhaps we might not describe them very clearly in the original manuscript. We have rephrased the statements in the revised manuscript as pointed out by this reviewer. Please allow us to give an additional explanation. The experiment is selected in the northeastern Tibetan Plateau in terms of the area at around 5.8 x $10^5$ km$^2$. To verify the performance of the new method, the original, bias-corrected and blended SPE are intercompared

at the random validation grid cells in the survey region (Lines 180-207). The time series of daily rainfall estimates and rainfall accumulations in terms of the original and blended SPE is further added at a selected validation location in the warm season of 2014 (Fig. 6; Lines 210-215). To mitigate the impact of validation locations, 10 randomly test is performed for the selection of validation grids (Lines 216-229). Also, a heavy rainfall event that occurred on September 22, 2014 is examined to quantify its performance in the extreme rainfall scenario (Lines 244-266). The two-step blending (TSB) method is also compared with the existing fusion algorithms (e.g., Bayesian model averaging (BMA) and One-outlier removed (OOR)) at the validation grids (Table 6; Lines 268-275). More details can be found in the revised manuscript.

Last but not at least, this manuscript needs to be further polish before publication.

**Response:** Polish as suggested.

**Anonymous Referee #2:**

This manuscript describes a two-step methodology to combine multiple satellite precipitation products to produce a blended daily precipitation estimate. The process involves first bias correcting the individual satellite QPE products relative surface rain gauges. Then, a Bayesian weighting is applied to blend the various QPE datasets into a single product. The approach is demonstrated on a small area in the northeastern part of the Tibetan Plateau over the 2014 warm season, as well as an individual heavy rain case. Overall the manuscript needs to be checked for correct grammar and usage, and the data and methods sections could be lengthened a bit to make things clearer and therefore reproducible (some specific suggestions for this below). Generally, with a few tweaks to the writing I feel this is publishable with minor revisions.

**Response:** We thank this reviewer for the great comments. The manuscript is carefully checked to avoid grammar and usage typos. The data and methods sections have been rephrased in the revised manuscript as required by this reviewer (Lines 68-174).

Specific Comments:

The manuscript would be much easier to follow if consistent terminology were used to refer to original SPE, bias corrected SPE, and blended SPE throughout.

**Response:** Revise as suggested.

Lines 75-80: Additional information about the data used is needed: Please specify the versions of IMERG and CMORPH you are using, and whether the IMERG is the near real time early, near real time late, or research/final runs. It is also interesting that you chose to use TMPA, which is no longer being produced and is generally very similar to IMERG. Additionally, IMERG, CMORPH, and TRMM-3B42 all have daily products available - why did you choose to use the 3-h products and (presumably) accumulate to daily? Finally, what method did you use to resample the IMERG?

**Response:** The CMORPH V1.0 research products and the Level 3 IMERG V03 final run products

are used in this study. We agree that TMPA is similar to IMERG, but the satellite retrieval algorithm between the two products are different. Considered that TMPA 3B42V7 shows a good performance in the TP, it is selected as an individual in this blending process. It is known that the daily scale of SPE is accumulated from the 3-h (TMPA, CMORPH) or 30-min (IMERG), we admit that we can directly use the daily scale instead of accumulation again from the 3-h products as suggested. The nearest neighbor interpolation is used to resampling the IMERG data. We have clarified these issues in the revised manuscript as pointed out by this reviewer (Lines 76-81).

Line 85: If you are using CMORPH V1.0, it also corrects using GPCP.

**Response:** Corrected as suggested.

Line 116: This equation would be easier to read if separated into 3 lines.

**Response:** Separated as suggested.

Line 162-170: Some discussion of the effects of comparing point data to somewhat low resolution gridded data is needed.

**Response:** We have rephrased this statement in the revised manuscript as suggested by this reviewer (Lines 86-88; Lines 106-107; Lines 279-284).

Line 182-183: It seems that the scatter is reduced for the blended product, but it has induced a high bias for low rain days and a low bias on heavy rain days. It's difficult to see if the bias is improved compared to the original SPE products.

**Response:** We thank this reviewer for the comment. Yes, the scatter is reduced for the blended SPE. We perform an additional comparison at the validation locations based on various rainfall intensities in Fig. 5 in the revision. Based on the TSB method, the blended SPE have been effectively dropped towards GR at the validation sites (Fig. 5b), especially for the rain intensity values less than 15 mm/d (Fig. 5c). Also, there is an overestimation for the original SPE but an underestimation for the blended SPE as the daily rainfall is more than 15 mm, partly because the BC process might over-correct the original SPE on the heavy rainfall in this case. Overall, this TSB method has its ability to exert benefits from SPE in terms of higher performances and mitigate poor impacts from the ones with lower quality. We have also rephrased this statement in the revised manuscript as pointed out by this reviewer (Lines 200-207).

[Figure]

*Figure 5: (a) The Box-Whisker plots of relative weights of the bias-corrected SPE in Stage 2; (b) Scatter plots between GR and various SPE (original and blended) at the validation grids in the warm season of 2014; (c) The PDF of daily rainfall in terms of the GR, original and blended SPE with various intensities at the validation grids in the warm season of 2014.*

Line 213: I disagree with this statement. PRECDR is clearly very different from the others, and to this point in the manuscript has shown very litter value to be kept in consideration, and I think it is worth acknowledging this, then using the case study to point out that PRECDR can in fact be informative and on a case by case basis.

**Response:** We fully agree with this reviewer for the comments. We have added the statements in the revised manuscript as pointed by this reviewer (Lines 205-207).

**Anonymous Referee #3:**

The manuscript by Ma et al. Presents a very interesting study on blending multiple satellite estimates to obtain a better precipitation estimates, especially over region with complex terrain. The analysis is systematic and results support the improvement in precipitation estimates due to two-stage blending approach. During my read, on several occasion I kept searching for necessary details. Unless those details are provided, it is hard to fully evaluate the merit of this work. Therefore I would suggest major revision of the current version of the manuscript.

**Response:** We thank this reviewer for the critical comments. More details of the TSB approach have been added in the revised manuscript as pointed out by this reviewer (Lines 96-174).

Authors may want to improve the manuscript along the following lines:

[1] Please provide full details of bias adjustment and data merging stages. With help of some example dataset, Authors need to describe how Equations [1], [2a] and [2b] adjust the bias. Similarly please demonstrate with some dataset how weight parameters were obtained from Equation [3].

**Response:** We thank this reviewer for the important comment. In the revised Method section, the process of bias adjustment and weight parameter estimation are explicitly described. The full details of bias adjustment and data merging stages are provided in the revised manuscript as suggested by this reviewer (Lines 96-174).

[2] Please include plots justifying why Student's t distribution was selected. I am sure at different training sites, different distributions (Lognormal, Gamma, etc.) may show better performance.

**Response:** We fully agree with this reviewer that at different training sites, different distributions (Lognormal, Gamma, etc.) may show better performance. Given various SPE at different training sites, the specific probabilistic function is not limited to a certain distribution. For demonstration purpose, we herein apply the Student's *t* distribution, with its mean parameter expressed as a linear regression of the original SPE and terrain feature in this case. The goodness-of-fit of the Student's t distribution for the bias between GR and SPE is examined graphically by using a quantile-quantile

plot at the training grids (Fig. 3). It is found that all of them are close to the diagonal red line. It indicates that the selection of Student's t distribution is basically acceptable. We have also rephrased this statement in the revised manuscript (Lines 114-119).

[Figure]

*Figure 3. Quantile-quantile plots at training grid cells for the bias between GR and SPE, where (a) to (d) shows PERCDR, 3B42V7, CMORPH, and IMERG, respectively.*

[3] Please explain how the information from gridded data (Satellite estimates) was transferred to point locations (training and validation) sites. Did Authors apply some downscaling approach? Bringing information from 25km grid to a point in a complex topographical region is challenging.

**Response:** We fully agree that it is challenging to bring information from 25 km grid to a point in a complex terrain region. Perhaps we didn't describe it very clearly in the original manuscript. The downscaling approach is not applied in this study. To ensure the same resolution among the original SPE, the IMERG data are resampled from 0.10° to 0.25° using the nearest neighbor interpolation to eliminate the scale difference. The rain gauge network is spatially interpolated with a 0.25° x 0.25° resolution in the region of interest on each rainy day using a bilinear interpolation approach. The 34 grid cells with the gauge sites are assumed as ground references (GR) in the blending process. We admit that there is a scale gap between SPE and gauge observations. We have clarified the statements and add some discussions in the revised manuscript as pointed out by this reviewer (Lines 79-88).

[4] In equation [1], normalized elevation is used as a covariate. If it is not included, how it will affect the result. Can you quantify it? Was that included just because you are dealing with TP? In the discussions (Section 5), Authors mention about the importance of including other covariates related to precipitation generation mechanism.

**Response:** We are thankful to this reviewer for the comment. We quantify the impact of elevation covariate on the bias-corrected and blended SPE performances as pointed out by this reviewer (Table 7). It is found that the consideration of elevation feature performs slightly better skill compared with

the model without terrain in this case study. We would like to admit that it is an initial exploration partly because we are dealing with the TP. We have rephrased the concerns in the revised manuscript as pointed out by this reviewer (Lines 279-284).

*Table 7. Statistical error indices (i.e., RMSE, NMAE, and CC) of the bias-corrected and blended SPE before (No Terrain) and after (Terrain) the consideration of terrain information at the validated grid locations in the warm season of 2014 over the NETP.*

| Product | Type | RMSE (mm/d) | NMAE (%) | CC |
|---------|------|-------------|----------|-----|
| BC-PER | No Terrain | 5.03 | 58.9 | 0.416 |
| | Terrain | 5.02 | 58.7 | 0.418 |
| BC-V7 | No Terrain | 5.08 | 58.0 | 0.403 |
| | Terrain | 5.06 | 57.5 | 0.410 |
| BC-CMO | No Terrain | 4.83 | 55.0 | 0.493 |
| | Terrain | 4.81 | 54.6 | 0.497 |
| BC-IME | No Terrain | 4.58 | 51.4 | 0.568 |
| | Terrain | 4.56 | 50.9 | 0.572 |
| Blended SPE | No Terrain | 4.36 | 49.7 | 0.603 |
| | Terrain | 4.34 | 49.2 | 0.606 |

[5] As mentioned in Section 2, the data of only warm period from May to September 2014 has been used in this study. Since all the satellite data are available for several years, can Authors perform similar analysis for few years and validate their approach?

**Response:** We thank the reviewer for this suggestion. Please allow us to explain that this study aims to develop a newly TSB algorithm on the multi-satellite precipitation data fusion in a certain time in regions of interest. Given that the larger challenge in the TP is to provide more accurate rainfall in a spatial domain, we are trying to overcome the shortage of limited rain gauge network based on the available SPE with spatial advantage using the TSB method in the NETP as a demonstration purpose. We agree that the satellite data are available for several years, but the exploration of long-term periods for the TSB method is another critical issue, e.g., the consideration of time impact on the fusion result.

Generally, the model performance of this new approach has been demonstrated based on various aspects in the revised manuscript. Please allow us to repeat them below: To verify the performance of the new method, the original, bias-corrected and blended SPE are intercompared at the random validation grid cells in the survey region (Lines 180-207). The time series of daily rainfall estimates and rainfall accumulations in terms of the original and blended SPE is further added at a selected validation location in the warm season of 2014 (Fig. 6; Lines 210-215). To mitigate the impact of validation locations, 10 randomly test is performed for the selection of validation grids (Lines 216-

229). Also, a heavy rainfall event that occurred on September 22, 2014 is examined to quantify its performance in the extreme rainfall scenario (Lines 244-266). The TSB method is also compared with the existing fusion algorithms (i.e., BMA and OOR) at the validation grids (Table 6; Lines 268-275). We thus consider that the evaluation analysis for long-term period and extended regions (e.g., TP) will be performed in a future study.

[6] Since similar approaches have been developed previously (as mentioned at the end of second paragraph of page 2, Authors should compare the results with the existing approach. The only unique feature of the current approach is that it provides predictive uncertainty.

**Response:** We thank the reviewer for this great suggestion. It is very important to formally quantify the predictive uncertainty in the Bayesian analysis, which is one of the unique features for the TSB method. In the revised manuscript, the TSB approach is compared with two existing fusion approach, i.e., BMA and OOR. The statistical summary of data comparison among the three fusion approaches at the validated locations are shown below. The TSB approach performs the best skill as compared with the other two fusion methods. We have added this comparison in the revised manuscript as kindly suggested by this reviewer (Lines 268-275).

*Table 6. Statistical error indices (i.e., RMSE, NMAE, and CC) of three blending approach (i.e., OOR, BMA, and TSB) at the validated grid locations in the warm season of 2014 over the NETP.*

| Method | RMSE (mm/d) | NMAE (%) | CC |
|--------|-------------|----------|-------|
| OOR | 5.63 | 59.2 | 0.547 |
| BMA | 5.44 | 57.6 | 0.595 |
| TSB | 4.34 | 49.2 | 0.606 |

[7] The results presented in Figures 3 and 4 homogenizes many things. In Figure 3, are you presenting the average value over all the validation sites? I am sure results will differ significantly if you look into individual sites. Also time series plots would show more features than the bar plot. The results from blended is similar to many adjusted SPE, then can it be concluded that there is no need to blend. Simply apply the stage 1, bias adjustment, and select the best SPE.

**Response:** We are thankful to the reviewer for these comments. Yes, Figure 3 presents the statistical error summary over all the validation grids. We agree that there are more features if looking into individual sites than overall evaluation of the validated sites. The time series plot of daily rainfall estimates and rainfall accumulations of GR, original and blended SPE at a validated grid cell with a rain gauge labeled as ID 56173 is shown in Figure 6 as a demonstration example in the revised manuscript. This rain gauge, which is located at (32.8° N, 102.55°E, 3484 m), has the maximum rainfall record in the warm season of 2014 in the NETP. Visual analysis shows that the blended SPE provides reasonable rainfall and has a better skill in terms of RMSE at 4.95 mm/d compared with the original SPE including PERCDR (10.71), 3B42V7 (9.76), CMORPH (8.0), and IMERG (10.49), respectively.

[Figure]

*Figure 6. Time series of daily rainfall estimates and rainfall accumulations at a selected validation grid with the maximum rainfall record in the warm season of 2014: (a) daily rainfall estimates, and (b) rainfall accumulations.*

This reviewer also raises a question that why not be careful at the first place in selecting a good set of SPE, or simply apply the first stage of bias correction and then select the best SPE as the final product. To address this issue, we investigate the error differences among the best-performed SPE, i.e., BC-IME, and blended SPE before and after the removing of the worst-performed bias-corrected SPE, i.e., BC-PER, for 10 random verified tests in the warm season of 2014 (Fig. 13). It shows that it is beneficial to involve the Stage 2 in the TSB method because the blended SPE performs better skill than BC-IME in the Stage 1 process. The primary reason is that the BW model is designed to integrate various types of bias-corrected SPE, which is limited in the BC model. Also, both blended SPE in Figure 13 show similar performances of the RMSE, NMAE, and CC indices. It implies that the TSB approach has an advantage of not impacted by the poor quality individuals (e.g., BC-PER), partly because the BW model can reallocate the contribution of the bias-corrected SPE based on their corresponding bias characteristics.

We have also rephrased the expressions in the revised manuscript as pointed out by this reviewer (Lines 200-215; Lines 286-295).

[Figure]

*Figure 13. Statistical error indices (i.e., RMSE, NMAE, and CC) of the best-performed bias-corrected SPE (i.e., BC-IME, black) and blended SPE before (red) and after (blue) the removing of the worst-performed BC-PER for 10 random tests in the warm season of 2014 in the NETP.*

[8] In Figure 4, Authors conclude that the blended data have been dropped towards the gauge references but please look at the precipitation with higher values. It appears that red dots have narrow spread for the lower values but SPE is over estimating the values.

**Response:** We are thankful to the reviewer for the important comment. We also notice that there is an overestimation for the original SPE compared to GR, and the blended SPE shows different spread at various rainfall intensities. To address this issue, we perform an additional analysis of probability density function of daily rainfall with various intensities at the validated locations blow (Fig. 5c). There is an overestimation for the original SPE but an underestimation for the blended SPE as the daily rainfall is more than 15 mm, partly because the BC process might over-correct the original SPE on the heavy rainfall in this case. We have rephrase this statement in the revised manuscript as pointed out by this reviewer (Lines 200-205).

[Figure]

*Figure 5*: *(a) The Box-Whisker plots of relative weights of the bias-corrected SPE in Stage 2; (b) Scatter plots between GR and various SPE (original and blended) at the validation grids in the warm season of 2014; (c) The PDF of daily rainfall in terms of the GR, original and blended SPE with various intensities at the validation grids in the warm season of 2014.*

[9] Authors claim that the two-stage approach has advantage of not getting impacted by the poor quality SPE. Based on Figure 4a, it can be argued that why to include those SPE which has very low weight. Please justify. Furthermore, Figure 6 shows improvement ratio, of course the SPE with very low weight will show high value here. Why not be careful at the first place in selecting a set of SPE?

**Response:** We thank this reviewer for the critical comments. It is well known that the SPE are obtained from different satellite retrieval algorithms, and each of them can provide various rainfall information. The over-performed SPE would provide more information, and the poor-performed ones give less value. It is thus necessary to integrate all kinds of SPE so as to reduce the predictive uncertainty in the domain. The proposed TSB approach has an advantage of integrating various SPE information and not impacted by the poor quality of SPE, partly because the BW model in Stage 2 can reallocate the contribution of the SPE based on their corresponding bias characteristics.

To address this issue, we also investigate the statistical error difference among the best-performed bias-corrected SPE (i.e., BC-IME), and blended SPE before and after the removing of the worst-performed bias-corrected SPE (i.e., BC-PER) in this case, for 10 random verified tests in the warm season of 2014 in the NETP (Fig. 13). It is found that the blended SPE performs better skill than the simply bias correction with BC-IME, and both blended SPE products show similar performances of the RMSE, NMAE, and CC indices. It proves that it is beneficial to involve the Stage 2 process in the TSB method.

We have rephrased the related expressions in the revised manuscript as pointed out by this reviewer (Lines 286-295).

[Figure]

*Figure 13. Statistical error indices (i.e., RMSE, NMAE, and CC) of the best-performed bias-corrected SPE (i.e., BC-IME, black) and blended SPE before (red) and after (blue) the removing of the worst-performed BC-PER for 10 random tests in the warm season of 2014 in the NETP.*

[10] Authors talk about CC for a rainfall event (Sept 22, 2014)? Given that the analysis is performed on daily data, how do you obtain CC?

**Response:** There are 27 rain gauge sites in total that has a rainfall record on September 22, 2014 in the regions of interest. The CC index is calculated based on the data sets from the 27 grid cells.

[11] Title says 'A flexible two-stage approach...' In the second paragraph of Section 6, Authors talk about what is the flexibility here. The statement is very general that it is capable of involving a group of multi-SPE. Is that so unique? Please look into it and accordingly modify the title.

**Response:** We thank this reviewer for the important comment. The word of "flexible" is removed in the title. We have replaced the title with "*A two-stage approach for blending multiple satellite precipitation estimates and rain gauge networks: An experiment in the northeastern Tibetan Plateau*" in the revised manuscript.

[12] Figure 8a is quite different from Figures 7a to 7d. By blending, higher values disappeared from the map except in Southwest corner. Please explain.

**Response:** Thank you for this specific comment. Figure 8a (i.e., Fig. 10a in the revised manuscript) is the spatial map of the blended result which is weight summation of the original SPE from Figures. 7a to 7d. There is an overestimation for most of the original SPE in the NETP in this experiment, the bias of the blended SPE is reduced based on the TSB approach. Thus, higher values disappear from the map except in southwest corner. Because daily mean rainfall is the highest in southwest corner for each SPE, higher value exists after the blending process. We have explained this issue in the revised manuscript as pointed out by this reviewer (Lines 238-242).

[13] The blending product will be extremely beneficial for the areas where there is no or very few rain gauges (specially in mountain area). However the study area was carefully selected in such a way that the rain gauge intensity is high. Can the results be extrapolated from the training and validations sites to get the improved blended gridded product, the way Authors have done in Figure 8? If yes, then there must be some guideline how many minimum training sites do I need to apply this two-stage approach in other complex regions.

**Response:** We are thankful to this reviewer for the comment. As pointed out by this reviewer, it is helpful to give some guideline that how many training sites are needed to apply the TSB approach in a region with complex terrain and limited GR. The sensitivity analysis of the number of training grid cells on the performance of blended SPE at the validated sites is explored in Figure 14. As the number of training sites is increasing, there is a decreasing trend for the RMSE and NMAE values, but a slight increasing trend for the CC value. Except for an anomaly with No. 23, the performance of the blended SPE becomes similar as the number of training sites increases to 21 in this case. Also, it is noted that if more useful information is provided from the involved SPE and rain gauges, it is more beneficial for the blended gridded product in the region of interest. We have rephrased this issue in the revised manuscript as pointed out by this reviewer (Lines 299-308).

[Figure]

*Figure 14. Statistical error indices (i.e., RMSE, NMAE, and CC) of the blended SPE at the validated grid locations in terms of different number of training sites in the warm season of 2014 in the NETP.*

[14] The manuscript should be thoroughly checked for grammar and usages.

**Response:** Thoroughly checked as suggested.

[revised manuscript text omitted]

**3.2 Stepage 1: bBias correction**

In this stepage, we focusperform on conditional modelling of GR on each of the original SPE, i.e., on the probabilistic distribution $f(R)$ at the training sets to improve the accuracy of the original SPE accuracy. With regard to the conditional distribution of GR, aA flexible assumption (e.g., Lognormal, Gaussian, or Student's $t$ distribution) for bias characteristics between GR and SPE is proposed. Given various SPE at different training sites, the specific probabilistic function is not limited to a certain distribution. For demonstration purposes, we apply the Student's $t$ distribution, with its mean parameter expressed as a linear regression of the original SPE. Also, Tthe goodness-of-fit of the Student's $t$ distribution for the bias between GR and SPE at the training grid cells is examined graphically by using a quantile-quantile plot at the training sets (Fig. 3). It is found that where they are close to the diagonal red line. A Student's $t$ distribution is thus adopted with its mean parameter expressed as a linear regression of SPE. It is

We parameteriszed the Student's $t$ distribution asas follows:

$$R \sim Student(v_i, \mu_i, \sigma_i) \quad (1)$$

$$\mu_i = \alpha_i + \beta_i * Y_i + \gamma_i * Z \quad (2)$$

where $v_i$ is known as degree of freedom, $\mu_i$ and $\sigma_i$ stand for sample mean and variance, respectively; the parameter $\mu_i$ is correlated with the intensity value of the $i^{th}$ SPE ($Y_i$) and associated terrain feature (e.g., elevation) (Z). To avoidignore the scale factordata anomaly, the normalized elevation feature ranging from 0 to 1 is used as the terrain feature in the regression model of Eq. (2). is normalized and its value ranges from 0 to 1 after the normalization. Also, $\theta = \{v_i, \alpha_i, \beta_i, \gamma_i, \sigma_i\}$ isis summarized as a the parameter set,s which enables to write the likelihood function or probability density function (PDF) from Eqs. (1) and (2) conditional on $\theta$ and $Y_i$ as:

$$f(R|\boldsymbol{\theta}, Y_i) = \frac{\Gamma((\nu_i+1)/2)}{\Gamma(\nu_i/2)} \frac{1}{\sqrt{\nu_i \pi}\,\sigma_i} \left(1 + \frac{1}{\nu_i}\left(\frac{R-(\alpha_i+\beta_i*Y_i+\gamma_i*Z)}{\sigma_i}\right)^2\right)^{-(\nu_i+1)/2} \tag{3}$$

According to the Bayes's theorem (Gelman et al., 2013), the posterior distribution of parameter set $\boldsymbol{\theta}$ given GR and SPE data, and the prior distribution of parameters $f(\boldsymbol{\theta})$ can be expressed as:

$$f(\boldsymbol{\theta}|R, Y_i) \propto f(R|\boldsymbol{\theta}, Y_i) f(\boldsymbol{\theta}) \tag{4}$$

150     The estimation of the posterior distribution $f(\boldsymbol{\theta}|R, Y_i)$ in Eq. (4) is challenging as its dimension grows with the number of parameters (Renard, 2011). Here, However, tthe Markov Chain Monte Carlo (MCMC) technique complied in the Stan programming language can beis used to address this issue (Gelman et al., 2013). ConsiderGivened that the assumption of the weakly informative priors ensures the Bayesian inference in an appropriate range (Ma et al., 2020b), the priors of $f(\boldsymbol{\theta})$ are initialized as uniform distribution with $\alpha_i, \beta_i, \gamma_i$ at real numbers in Eq. (5), and with $\nu_i, \sigma_i$ at a lower-bound zero of real 155     numbers in Eq. (6).

$$\alpha_i, \beta_i, \gamma_i \sim Uniform(-\infty, +\infty) \tag{5}$$

$$\nu_i, \sigma_i \sim Uniform(0, +\infty) \tag{6}$$

Based on the estimated parameter set $\boldsymbol{\theta}$ above, the next step is to calculate each of the bias-corrected SPE $R^*$ at any new site of the domain at the same period,. whichIt can-can be quantitatively simulated from its posterior predictive distribution in Eq. 160     (7) using the associated original SPE $Y_i^*$, and training data $R, Y_i$:

$$f(R^*|Y_i^*, R, Y_i) = \int f(R^*, \boldsymbol{\theta}|Y_i^*, R, Y_i)\, d\boldsymbol{\theta} \tag{7}$$

[revised manuscript text omitted]

**4 Results**

To assess the performance of the proposed two-stage blendingTSB method, several statistical error indices including Rroot Mmean Ssquare Eerrors (RMSE), nNormalized Mmean Aabsolute Eerrors (NMAE), and the Pearson's Ccorrelation Ccoefficients (CC) are used in this study. The specific formulas of these metrics can be found in the literature (e.g., Chen et al., (2019 among others).

**4.1 Evaluation of the original, Bbias-corrected, and blended adjustment of multi-SPE at the validationed grids**

Compared to the the gauge referencesGR, the original multi-SPE including (i.e., PERCDR, 3B42V7, CMORPH and IMERG) show significant large biases at the independent validation grid sites cellsgrids over in the NETP during the warm season of 2014 (Table 2). Their statistical error metrics including RMSE, NMAE, and CC range from 6.59-8.07 mm/d, 63.2-83.5%, and 0.403-0.5768, in terms of RMSE, NMAE, and CC, respectively. 3B42V7 performs has the worst skill with the highest RMSE of 8.07 mm/d and the highest NMAE at of 83.5%, and the lowest CC of 0.403. IMERG shows the best performance in terms of the lowest NMAE atof 63.2% and highest CC at 0.5768 among the four SPE., which presents its superiority compared with the other SPE in the survey area. It seems that the satellite retrievals need to be further clarified with regard to the mainstream SPE in the NETP.

After the bias Based on the BC modeladjustment of each SPE, the updated multi-bias-corrected SPE (i.e., BC-PER, BC-V7, BC-CMO and BC-IME) show great improvement have better agreements with GR in data qualityat the validation grids in the

[revised manuscript text omitted]

---

## Author Response (AR2)

**Responses to Review Comments on HESS-2020-43-R1**

We thank the anonymous referee #3 for the comments and suggestions. We have addressed the comments point-by-point in the revision. In the text below, comments are repeated verbatim and the corresponding responses are in blue. Also, we have made substantial improvements to the manuscript based on the comments and suggestions, and the label lines in brackets below are based on the clean version of the revised manuscript.

**Anonymous Referee #3**

Authors have included some of my comments in the revised version of the manuscript. However, it is clear that Authors are resisting to look at some of the comments which actually raise serious objections on the efficacy of the proposed two-steps blending approach and consequently the validation of results. In the previous version also, I had commented that details were not provided on many critical aspects. Authors have admitted the limitations of the previous version of the manuscript. Now in the revised version, they have somehow tried to satisfy my comments by adding more analysis but again the presented details in the methodology are hardly sufficient. In summary, I feel resistant from Authors to perform any further investigation about their approach, analysis with more years of data and provide more details on the parameter estimation using a dataset. In my view, the revised manuscript tries to demonstrate a potential approach with lack of in-depth analysis. I would recommend major revision.

**Response**: We appreciate the reviewer for the critical comments. In this revision, the approach is further investigated and clarified as required by this reviewer. Also, we perform more years of data in the warm season of 2010 to 2014 for model validation and provides more details of parameter estimation. Detailed information can be found in the following point-by-point responses and the revised manuscript.

Authors may want to work on my following comments:
[1] Section 3.2 says "The goodness-of-fit of the Student's t distribution for the bias between GR and SPE is examined graphically by using a quantile-quantile plot at the training sets (Fig. 3). It is found that they are close to the diagonal red line." Please look at Figure 3, the distribution is

completely skewed. Blue dots are hardly lying on the diagonal red line. If the distribution is not properly set, then the parameter estimation and results will have errors.

**Response**: We thank the reviewer for the important comment. We have fixed the mistake in the revised version. Student distribution is a symmetric distribution, which is therefore not suitable for skewed data. To find a suitable distribution, we tested several distributions including Lognormal, Gaussian (just for comparison) and Gamma distribution. It is found that a Gamma distribution is more appropriate as its Probability-Probability (PP) plots are closer to the diagonal black line for the training data in the warm season of 2014 in the northeast Tibetan Plateau (Figure 3). In the revision, a Gamma distribution is replaced in the first stage. We have also rephrased the related expression in the revised manuscript as pointed out by this reviewer (Lines 107-111).

[Figure]

**Figure 3**: (a) The histogram density plot and (b) the corresponding Probability-Probability plot of GR at the training grids in the warm season of 2014 in the NETP, where the red, blue and green lines shows the fitted Gamma, Lognormal and Gaussian distribution, respectively..

Regarding the estimation of parameters, please look at Robertson et al. 2013, 'Post-processing rainfall forecasts from numerical weather prediction models for short-term streamflow forecasting' where MCMC approach of estimating parameters were changed to MAP approach based on the available length of data. Also please look at the parameter estimation section in Shrestha et al. 2015, 'Improving Precipitation Forecasts by Generating Ensembles through Postprocessing.' This is one of the reasons why I had requested Authors to explain the steps to estimate parameters using

some example dataset. It seems that Authors are hesitating to present this in the manuscript. Instead, Authors have added few more equations in the methodology section to make it appear more descriptive.

**Response**: We appreciate for the reviewer's suggestion, especially the two papers. We have added the two references in the revised manuscript. We also agree that accurate parameter information is fundamental for model inference as pointed out by this reviewer.

In the revised manuscript, the Markov Chain Monte Carlo (MCMC) technique with its sampling algorithm as the No-U-Turn Sampler (NUTS) variant of Hamiltonian Monte Carlo in the Stan program is performed to address this issue. The sampling records of model parameters are obtained based on the training data in the warm season of 2014 in the NETP. Since we only have four parameters in this model, the MCMC converges very quickly. Thus, we run a chain of length 2000, removing the first 1000 iterations as the warm-up period and retaining the second 1000 iterations. The parameter samples of these 1000 iterations are the samples of the posterior distribution $f(\theta_i|\mathbf{R}, Y_i, \mathbf{Z})$. The prediction of bias-corrected SPE in Stage 1 is performed using the MCMC iterated samplings. As for each SPE, a numerical algorithm is suggested based on the replicate of the post-convergence MCMC samples. The mean value of the MCMC samples $R_k^*$, denoted by $Y_i'$, is regarded as the bias-corrected SPE and the associated credible intervals (e.g., 2.5% and 97.5% quantiles) is used for predictive uncertainty. In Step 2, the estimation process in a Bayesian framework is similar to that described in Stage 1. After all parameters are estimated, as similar to the Bayesian inference in Stage 1, the blended SPE at any site and time can be derived with the bias-corrected SPE and corresponding weights using the MCMC iterations.

In the revised manuscript, the TSB model structure, parameter estimation and Bayesian inference are reorganized in "*Section 3.1 TSB*" of the "*Methodology*" part (Lines 94-185). In addition, the parameter estimates are analyzed in "*Section 4.1 Parameter estimates*" of the "*Result*" part (Lines 226-237), where Figures 4 and 5 shown below are the PDF curves of posterior parameter sets in the bias-correction and data merging stages.

[Figure]

**Figure 4**: The PDF curves of posterior parameter sets with regard to (a) PERCDR, (b) 3B42V7, (c) CMORPH and (d) IMERG in the bias correction process of Stage 1.

[Figure]

**Figure 5:** The PDF curves of posterior parameter sets in the data fusion process of Stage 2.

[2] In response to my comments, now in Section 2, it is mentioned that "The rain gauge data are spatially interpolated with a 0.25° x 0.25° resolution in the study region for each rainy day using

a bilinear interpolation approach. The 34 grid cells with the gauge sites are assumed as ground references (GR) in the blending process." This poses a serious limitation on the analysis. Given the complexity of the region, a simple bilinear interpolation approach is hard to justify. Look at figure 1, the elevation changes from 785 to 6252. The rain gauges stations are also far from each other, they are not dense.

**Response**: We thank the reviewer for this critical comment. We admit that the rain gauge stations are not very dense in the study area, but a denser ground network is not available in the short time. To address this significant concern, the China Gauge-based Daily Precipitation Analysis (CGDPA) at 0.25° and daily resolutions is replaced as the source of ground information in this study. The CGDPA is developed with a dense gauge network including 2400 rain gauges in mainland China using a climatology-based optimal interpolation and topographic correction algorithms (Shen and Xiong, 2014). The 34 grid cells with the gauge sites are assumed as ground references (GR) in the blending process. We have also rephrased the relevant statement in the revised manuscript (Lines 86-92).

*Shen, Y. and Xiong, A.: Validation and comparison of a new gauge-based precipitation analysis over mainland China. Int. J. Climatol., 36, 252-265, 2016.*

[3] In response to my comments, Authors wrote that "This study aims to develop a newly TSB algorithm on the multi-satellite precipitation data fusion in a certain time in regions of interest. Given that the larger challenge in the TP is to provide more accurate rainfall in a spatial domain, we are trying to overcome the shortage of limited rain gauge network based on the available SPE with spatial advantage using the TSB method in the NETP as a demonstration purpose. We agree that the satellite data are available for several years, but the exploration of long-term periods for the TSB method is another critical issue, e.g., the consideration of time impact on the fusion result." This response is hardly justified because Authors can repeat the validation for other years, especially when datasets are online available. Without this, I would have low confidence in most of the discussion in the result section.

**Response**: In the revision, the model performance is also validated with other years (2010-2013) as required by this reviewer. To address the reviewer's concern, model validation is performed under two scenarios: Scenario 1 will validate the model in space based on the data of the same period in validation stations (i.e., the 7 red grids in Figure 1), and Scenario 2 will validate the model in time based on the data of warm season from 2010 to 2013 at the same 27 black grids in

Figure 1. In addition, we consider a 10-fold cross validation in space by randomly selecting 7 sites for model validation, and the data of the remaining 27 sites as the training set. The performance of TSB approach is further compared with BMA and OOR in the two scenarios.

The model justification of the TSB approach are rephrased in the "*Section 4.2 Model validation under two scenarios*", "*Section 4.3 Cross-validation*", "*Section 4.4 Model comparison with BMA and OOR*". Herein, Table 2 shows the summary of statistical error indices (i.e., RMSE, NMAE, and CC) of the original SPE (PERCDR, 3B42V7, CMORPH and IMERG), bias-corrected SPE (BC-PER, BC-V7, BC-CMO and BC-IME), and blended SPE under two scenarios. More details can be found in Lines 239-311 in the revised manuscript.

**Table 2**: Summary of statistical error indices (i.e., RMSE, NMAE, and CC) of the original, bias-corrected and blended SPE in two scenarios in the NETP.

| Scenarios | Product | RMSE (mm/d) | NMAE (%) | CC |
|---|---|---|---|---|
| Scenario 1 | PERCDR | 7.15 | 70.2 | 0.382 |
| | 3B42V7 | 8.56 | 80.3 | 0.383 |
| | CMORPH | 6.25 | 60.6 | 0.556 |
| | IMERG | 6.60 | 62.9 | 0.506 |
| | BC-PER | 6.00 | 63.5 | 0.346 |
| | BC-V7 | 5.83 | 61.4 | 0.408 |
| | BC-CMO | 5.43 | 56.3 | 0.533 |
| | BC-IME | 5.44 | 56.0 | 0.530 |
| | Blended SPE | 5.36 | 54.6 | 0.570 |
| Scenario 2 | PERCDR | 9.19 | 79.3 | 0.261 |
| | 3B42V7 | 8.38 | 71.3 | 0.403 |
| | CMORPH | 7.20 | 61.9 | 0.493 |
| | IMERG | 7.64 | 65.1 | 0.462 |
| | BC-PER | 7.03 | 64.5 | 0.253 |
| | BC-V7 | 6.69 | 61.3 | 0.395 |
| | BC-CMO | 6.41 | 58.2 | 0.480 |
| | BC-IME | 6.44 | 57.7 | 0.470 |
| | Blended SPE | 6.37 | 56.7 | 0.513 |

[4] Authors have tried to satisfy my comments by adding a small section on comparing the proposed approach with existing BMA and OOR approaches. There are no details on BMA and OOR. It is entirely left up to the readers to figure out all these from previous literature.

I am repeatedly asking for details because the research topic which Authors are trying to address in this manuscript is extremely challenging. If the selection of distribution, parameter estimation etc. have known drawbacks then demonstrating better values of RMSE, MAE and CC does not prove the efficacy of the TSB approach.

**Response**: We are thankful for this kind suggestion. The method details of BMA and ORR have been added in the "*Sections 3.2 Comparison Model*" in the revised manuscript as requested by this reviewer (Lines 186-209). In this study, the BMA approach makes use of four original satellite data and the corresponding GR data at the 27 black grids shown in Figure 1 in the warm season of 2014 to estimate the optimal BMA weights. In Scenario 1, the BMA data are calculated based on the BMA weights and the original SPE from the 7 red grids in the warm season of 2014, and the OOR data are calculated based on the OOR method using the original SPE data from the 7 red grids in the warm season of 2014. In Scenario 2, the BMA data are calculated based on the BMA weights and the original SPE from the 27 black grids in the warm season from 2010 to 2013, and the OOR result are calculated based on the OOR method and the original SPE data from the 27 black grids in the warm season from 2010 to 2013.

The description of BMA and OOR is repeated below:

*"3.2 Comparison Model*

*3.2.1 BMA*

*The BMA method is a statistical algorithm that merges predictive ensembles based on the individual SPE at the training period in regions of interest. Here, the BMA result refers to the ensemble SPE. Based on the law of total probability, the conditional probability of the BMA data on the individual SPE is expressed as:*

$$f(BMA|Y_1, \dots, Y_p) = \sum_{i=1}^{p} f(BMA|Y_i) \cdot w_i \tag{17}$$

*where $f(BMA|Y_i)$ is the predictive PDF given by the individual SPE $Y_i$ and $w_i$ is the corresponding weight. The log-likelihood function l is applied to calculate the BMA parameter set $\vartheta$, which is written as:*

$$l(\vartheta) = log\left(\sum_{i=1}^{p} w_i \times f(BMA|Y_i)\right) \tag{18}$$

*It is assumed that $f(BMA|Y_i)$ follows a Gaussian distribution with its parameters as $\theta_i$, and BMA is ideally close to GR at any site and time. Eq. (18) is written as:*

$$l(\vartheta) = log(\sum_{i=1}^{p} w_i \times g(GR|\theta_i))\tag{19}$$

*where $g(\cdot)$ stands for Gaussian distribution, and $\vartheta = \{w_i, \theta_i, i = 1, ..., p\}$. The optimal BMA parameters $\vartheta$ are calculated by maximizing the log likelihood function using the expectation–maximization algorithm. In this study, the training period is set as the warm season of 2014 in the NETP. Before the execution of 
[revised manuscript text omitted]

$$f(R^*, \boldsymbol{\theta}_i\boldsymbol{\theta}|Y_i^*, Z_i^*, \boldsymbol{R}, \boldsymbol{Y}_i, \boldsymbol{Z}) = f(R^*|Y_i^*, Z_i^*, \boldsymbol{R}, \boldsymbol{Y}_i, \boldsymbol{Z}, \boldsymbol{\theta}_i\boldsymbol{\theta})f(\boldsymbol{\theta}_i\boldsymbol{\theta}|Y_i^*, Z_i^*, \boldsymbol{R}, \boldsymbol{Y}_i, \boldsymbol{Z}Y_i^*, \boldsymbol{R}, \boldsymbol{Y}_i) \tag{89}$$

Given that the new bias-corrected SPE $R^*$  is independent  to the training data, the first term of the right side in Eq. (9) is transformed as:

200
$$f(R^*|Y_i^*, Z_i^*, R, Y_i, Z, \theta_i \text{}, \theta) = f(R^*|Y_i^*, Z_i^*, \theta_i \theta) \qquad (\text{}(9\underline{10})$$

Since the parameters $\theta_i \theta$ are only dependent upon the training data $R, Y_i, Z,$ the second term of the right side in Eq. (9) is expressed as:

$$f(\theta_i|Y_i^*, Z_i^*, R, Y_i, Z)(\theta|Y_i^*, R, Y_i) = f(\theta_i \theta|R, Y_i, Z) \qquad (\text{}11)$$

Therefore, the  predictive  PDF of $R^*$ in Eq. (8) is written below:

205
$$f(R^*|Y_i^*, Z_i^*, R, Y_i, Z \text{}) = \int f(R^*|Y_i^*, Z_i^*, \theta_i \text{}, \theta) f(\theta_i|R, Y_i, Z)(\theta|R, Y_i) \, d\theta_i \theta \qquad (\text{}12)$$

Since there is no general way to calculate the associated integral in Eq. (12), the prediction is performed  using the MCMC iterated samplings (Renard, 2011). As for each SPE, a numerical algorithm is suggested below, where $n_{sim}$ stands for the replicate of the post-convergence MCMC samples and is set as 1000 in the case study. Thus, the predicted samples for $R^*$ in Eq. (12) are iterated ($k$ = 1 $n_{sim}$) as follows:

1) For the $i^{th}$ satellite product, randomly select a parameter sample $\theta_i \theta = \{\alpha_i, \delta_i, \beta_i, \gamma_i\}$ from the MCMC samples ;

2) Generate a value $R_k^* R^*$ from a  $Gamma\left(\alpha_i, \frac{\alpha_i}{\mu_i^*}\right)$ ., where $\log(\mu_i^*) = \delta_i + \beta_i * Y_i^* + \gamma_i * Z^*$;

Repeating step 1 and 2 for $n_{sim}$ times, the samples $R_k^*$ ($k = 1:n_{sim}$) are regarded as the realizations of the distribution of the bias-corrected SPE associated to the satellite estimation $Y_i^*$ and normalized elevation $Z^*$. The mean value of the samples $R_k^*$, denoted by $Y_i'$, is regarded as the bias-corrected SPE and the associated credible intervals (e.g., 2.5% and 97.5% quantiles) is used for predictive uncertainty.

220

**3.1.3 2 Stage 2: Data merging**

Ideally, the blended SPE ($B$) should be close to GR, i.e., $R$. Given the Gamma distribution of GR in Step 1, the blended SPE can be parameterized below: In Stage 1, the median value of the posterior samples is used as the bias-corrected SPE. Here, we redefine the bias-corrected SPE as $Y_t^{t}$ ($i = 1,2,\dots,p$). The formulas of blending the bias-corrected SPE are shown below:

225

$$B \sim Gamma\left(\alpha_B, \frac{\alpha_B}{\mu_B}\right) \tag{13}$$

where $\alpha_B, \mu_B$ and $\frac{\alpha_B}{\mu_B}$ are the shape, mean and rate parameters, respectively. In this step, the bias-corrected SPE of 4 satellites are merged with weight parameters $w_i$ ($i = 1,\dots,4$), and $\varepsilon$ is the residual error. The data fusion of bias-corrected SPE specified in the $log$ scale is defined as follows:

230

$$\log(\mu_B B) = \sum_{i=1}^{p4} \log(Y_i') * w_i + \varepsilon \tag{1214}$$

$$\sum_{i=1}^{4} w_i = 1 \tag{15}$$

$$\sum_{i=1}^{p} w_i = 1 \tag{13}$$

$$\varepsilon \sim Normal(0, \sigma_\varepsilon \sigma) \tag{1416}$$

235

$$w_i \sim Uniform(0,1), i = 1,\dots,p \tag{15}$$

$$\sigma \sim Uniform(0, +\infty) \tag{16}$$

where $B$ is the blended SPE; $w_i$ ($i=1,2,\dots,p$) stands for the relative weight of the $i^{th}$ bias-corrected SPE; $\varepsilon$ is the residual error. Ideally, the blended SPE at the training site $s$ and time $t$ should be close to GR, i.e., $R(s, t)$. Thereby, model all parameters $\delta$,

240 including $\alpha_B, w_i$ ($i = 1,2,\dots p4$) and $\sigma_\varepsilon \sigma$ will can be estimated based on from the GR and bias-corrected SPE at the training sites. With regard to the conditional distribution of blended SPE on the bias-corrected SPE, we propose a Gaussian distribution for the residual error modelling. The corresponding PDF is written as follows:

$$\cancel{f(B|\delta) = \frac{1}{\sqrt{2\pi}\sigma}\exp(-\frac{1}{2}(\frac{B-\sum_{i=1}^{p}Y_i^t*w_t}{\sigma})^2)} \tag{17}$$

The   process  $\delta$  is similar to th  described in

245    the Stage 1. After all parameters are estimated,    the blended SPE at any site and time  can be derived with the bias-corrected SPE and corresponding weights using the MCMC iterations.

250

**3.2 Comparison model**

**3.2.1 BMA**

The BMA method is a statistical algorithm that merges predictive ensembles based on the individual SPE at the training period in regions of interest. Here, the BMA result refers to the ensemble SPE. Based on the law of total probability, the conditional

255    probability of the BMA data on the individual SPE is expressed as:

$$f(BMA|Y_1,\dots,Y_p) = \sum_{i=1}^{p} f(BMA|Y_i) \cdot w_i \tag{17}$$

where $f(BMA|Y_i)$ is the predictive PDF given by the individual SPE $Y_i$ and $w_i$ is the corresponding weight. The log-likelihood function $l$ is applied to calculate the BMA parameter set $\vartheta$, which is written as:

$$l(\vartheta) = \log\left(\sum_{i=1}^{p} w_i \times f(BMA|Y_i)\right) \tag{18}$$

260    It is assumed that $f(BMA|Y_i)$ follows a Gaussian distribution with its parameters as $\theta_i$, and BMA is ideally close to GR at any site and time. Eq. (18) is written as:

$$\cancel{\log}\ \cancel{\log}\,l(\vartheta) = \log(\sum_{i=1}^{p} w_i \times g(GR|\theta_i)) \tag{19}$$

where $g(\cdot)$ stands for Gaussian distribution, and $\vartheta = \{w_i, \theta_i, i = 1,\dots,p\}$. The optimal BMA parameters $\vartheta$ are calculated by maximizing the log likelihood function using the expectation–maximization algorithm. Before executing the BMA method,

265    both GR and SPE data are pre-processed using the Box-Cox transformation to ensure that $f(BMA|Y_i)$ $(i = 1,\dots,4)$ is close to Gaussian distribution. As the BMA weights, $w_i, i = 1,\dots,4$ are obtained, the BMA data is calculated by weighted sum of the original SPE at any site and time. More details of the BMA method can be found in Ma et al. (2018).

**3.2.2 OOR**

270    The OOR method is defined as the arithmetic mean of the individual SPE by removing the feature with the largest offset. It is written as:

$$OOR = \frac{1}{p-1}\sum_{i=1}^{p-1} Y_i \tag{20}$$

where $Y_i$ is the individual SPE, $p$ is the number of SPE. The original SPE with the largest offset among the satellite products is removed and the average of the remaining SPE is regarded as the OOR result.

275    ### 3.3 Error analysis

To assess the performance of the proposed TSB method, several statistical error indices including root mean square errors (RMSE), normalized mean absolute errors (NMAE), and the Pearson's correlation coefficients (CC) are used in this study. The specific formulas of these metrics can be found  below:

$$RMSE = \sqrt{< (Sim - Obs)^2 >} \tag{21}$$

280
$$NMAE = \frac{<|Sim-Obs|>}{<Obs>} \times 100\% \tag{22}$$

$$CC = \frac{\sum[(Sim-<Sim>)(Obs-<Obs>)]}{\sqrt{\sum(Sim-<Sim>)^2}\sqrt{\sum(Obs-<Obs>)^2}} \tag{23}$$

where $Sim$ and $Obs$ stand for the simulated and observed data, respectively; the angle brackets stand for sample average.

**4 Results**

285    In the experiment, model parameters are calibrated on the daily precipitation of warm season in 2014, where GR data at the 27 black grids in Figure 1 are randomly selected for training the model. The model validation is performed under two scenarios: Scenario 1 will validate the model in space based on the data of the same period in validation stations (i.e., the 7 red grids in Figure 1), and Scenario 2 will validate the model in time based on the data of warm season from 2010 to 2013 at the same 27 black grids in Figure 1. In addition, we consider a 10-fold cross validation in space by randomly selecting 7 sites for model

290 validation, and the data of the remaining 27 sites as the training set. The performance of TSB approach is further compared with BMA and OOR in the two scenarios.

**4.1 Parameter estimates**

Figures 4 and 5a to 4d shows the posterior distribution curves of the posterior parameters in Stage 1 and 2, respectively. As
for each parameter in the bias-corrected process, the individual SPE including PERCDR, 3B42V7, CMORPH and IMERG
295 shows similar PDF pattern (Figs. 4a to 4d). in terms PERCDR, 3B42V7, CMORPH and IMERR in this step, respectively.
For each parameter, the individual SPE shows similar PDF curve. It showsseems that the bias structures of the original SPE
have similar characteristics. For all SPE, the distribution mass of parameter $\beta_i$ are all on the right side of zero, which implies
that a systematic bias exists for all satellite productsSPE. When looking at the effects of In addition, the elevation, the posterior
distribution of parameter $\gamma_i$ for PERCDR, 3B42V7 and CMORPH (Figs. 4a, 4b and 4c) have value zero in the middle range
300 of the distribution, which implies that elevation may have little impacts on these three satellite products. While forranges from
$-0.5$ IMERG in Fig. 4d, the distribution mass of parameter $\gamma_i$ is mostly on the right side of zero, which implies a clear effect
of elevation on this satellite product. to 0.5 among the satellite products, where the PDF pattern is similar between 3B42V7
and IMERG. It implies that the effect of elevation feature on the bias-corrected SPE has similar performance for 3B42V7 and
IMERG. In the data mergingfusion step (Fig. 5), IMERG has the highest weight and PERCDR has the lowest weight among
305 the four bias-corrected SPE. Moreover, 3B42V7 and PERCDR have similar contribution on the blended result (Fig. 5).
Basically, the Bayesian analysis is able to simulate the parameter uncertainty as as compared with the traditionally statistical
method. Figure 5 displays the PDF curves of the inferred posterior parameters in this step. It can be seen that the IMERG
product has the highest weight and PERCDR has the lowest weight among the four bias-corrected SPE.

310 To assess the performance of the proposed TSB method, several statistical error indices including root mean square errors
(RMSE), normalized mean absolute errors (NMAE), and the Pearson's correlation coefficients (CC) are used in this study.
The specific formulas of these metrics can be found in the literature (e.g., Chen et al., 2019 among others).

**4.1 2 Evaluation of the original, bias-corrected, and blended SPEModel validation under two scenarios**

315

Table 2 presents the summary of the statistical error indices including RMSE, NMAE and CC of the original (i.e., PERCDR,
3B42V7, CMORPH and IMERG), bias-corrected (i.e., BC-PER, BC-V7, BC-CMO and BC-IME) and blended SPE under two

scenarios in the NETP. The sub-section 4.2.1 and 4.2.2 shows the performance of the model validation under Scenario 1 and 2, respectively.

**320 4.2.1 Scenario 1**

 at the validation grids

the original SPE show large biases the RMSE, NMAE, and CC indices range from 6.59-8.073026.538-0.55(Table 2)56and.5%,3IMERGin terms of0.6at68survey area~~
[revised manuscript text omitted]
 me~di~an value at three gauge-based ~sit~grides ~during~on a heavy rainfall case ~on~of September 22, 2014: (a) ID 56171, (b) ID 561~52~73~,~ and (c) ID 56~18~2067. The original SPE and GR at each pixel are also indicated in each subfigure.

**Figure 11:** Spatial patterns of the daily mean precipitation in terms of the original SPE in the warm season of 2010 to 2014 in the NETP: (a) PERCDR, (b) 3B42V7, (c) CMORPH, and (d) IMERG.

715 **Figure 12:** Spatial patterns of the blended SPE in terms of (a) mean, (b) lower quantile (2.5%) and (c) upper quantile (97.5%) of daily mean precipitation in the warm season of 2010 to 2014 in the NETP.

**Figure 13.** Statistical error indices (i.e., RMSE, NMAE, and CC) of the best-performed bias-corrected SPE (i.e., BC-IME, black) and blended SPE before (red) and after (blue) removing the worst-performed BC-PER at 10 random verified tests ~for~

720 ~10 random verified tests~ in the warm season of 2014 in the NETP.

**Figure 14:** Statistical error indices (i.e., RMSE, NMAE, and CC) of the blended SPE at the validation grid locations in terms of different number of training sites in the warm season of 2014 in the NETP.

**Table 1:** Basic information of the original SPE used in this study.

| Short name | Full name and details | Temporal resolution | Spatial resolution | Input data | Retrieval algorithm | References |
|---|---|---|---|---|---|---|
| PERCDR | Precipitation Estimation from Remotely Sensed Information using Artificial Neural Networks (PERSIANN) Climate Data Record (CDR) | Daily | 0.25° x 0.25° | Warm season from 2014.50 to 2014.9 | Adaptive artificial neural network | *Ashouri et al., 2015* |
| 3B42V7 | TRMM Multi-satellite Precipitation Analysis (TMPA) 3B42 Version 7 | 3 hourlyDaily | 0.25° x 0.25° | Warm season from 2010 to 2014 2014.5-2014.9 | GPCC monthly gauge observation to correct this bias of 3B42RT | *Huffman et al., 2007* |
| CMORPH | NOAA Climate Prediction Centre (CPC) Climate Prediction Center (CPC) MorphORPHing tTechnique (CMORPH) for bias-corrected product version 1.0Global Precipitation Estimates Version 1 | 3 hourlyDaily | 0.25° x 0.25° | Warm season from 2010 to 2014 2014.5-2014.9 | Morphing technique | *XieJoyce et al., 20041 7* |
| IMERG | Integrated Multi-satellitE Retrievals for the Global Precipitation Measurement (GPM) mission V036 Level 3 final run product | 0.5 hourlyDaily | 0.10° x 0.10° | Warm season from 20102014.5-to 2014.9 | 20147 version of the Goddard profiling algorithm | *Huffman et al., 2018* |

725

**Table 2:** Summary of statistical error indices (i.e., RMSE, NMAE, and CC) of the original,  and blended SPE  in two scenarios in the NETP.

[revised manuscript text omitted]

---

## Author Response (AR3)

**Response to Review Comments on HESS-2020-43**

We thank this anonymous referee for the comments. We have addressed the comments point-by-point in the revision. In the text below, comments are repeated verbatim and the corresponding responses are in blue. Also, we have made necessary improvements to the manuscript.

Reviewer comments:

1.  Authors have modified the manuscript with most of my comments. Although my request to demonstrate the estimation of model parameters is still pending. This can be easily done in an appendix or supplementary material. Instead, Authors have elaborated on the steps involved in parameter estimation which was very much required.

**Reply:** We thank the reviewer for the comments. It is noted that the estimation of model parameters is described in Section 3.1. In our model, the calculation of likelihood is straightforward based on the equations described in Section 3.1, and parameter estimation can be easily achieved using any classical MCMC methods. This is really not an issue compared with the development of the two-stage blending approach. Because we want to fully express the proposed two-stage blending approach in the Section 3 Methodology part, we didn't separately extract the material of parameter estimation in an appendix or supplementary material in the manuscript.

2.  Since Authors have now made it clear that the precipitation estimate follows a gamma distribution (Eq 1 in the revised manuscript), isn't it important then to describe how the same (gamma) can be used in BMA without converting it to a normal distribution? Don't we need a Box-cox or log-sinh transformation to transform the data before applying BMS? Please clarify.

**Reply:** The BMA method used in this study is referred from Ma et al (2018)-JGR, where the training data are preprocessed using the Box-Cox transformation prior the BMA approach to ensure its Gaussian distribution in the merging process. Here, the purpose is to compare the proposed two-stage blending method with the existing BMA method, but not to revise the existing BMA method. It is interesting to examine the performance of the same (gamma) distribution in BMA without converting it to a normal distribution. However, it is out of the scope in this study. We have added this perspective in Section 4.4. In terms of whether we need a Box-cox or log-sinh transformation to transform the data before applying BMA, this depends on whether the proposed distribution can fit the data or not. Usually, a Gaussian distribution is suitable for the data after making Box-cox or log-sinh transformation, because the skewness of the data is reduced. However, an asymmetric distribution may not be suitable for the data after making Box-cox or log-sinh transformation. Here, because the gamma distribution shows a satisfying PP plots for the training data in this study, it is not necessary to perform a Box-Cox or log-sinh transformation to reduce the skewness of the data before applying BMA.

[revised manuscript text omitted]